# Multiplication-Free Parallelizable Spiking Neurons with Efficient Spatio-Temporal Dynamics

**Peng Xue**[1,2,6] **Wei Fang**[3]* **Zhengyu Ma**[1] **Zihan Huang**[4] **Zhaokun Zhou**[1,3]
**Yonghong Tian**[1,3,4] **Timothée Masquelier**[5] **Huihui Zhou**[1]*

[1]Peng Cheng Laboratory
[2]Shenzhen Institute of Advanced Technology, Chinese Academy of Sciences
[3]School of Electronic and Computer Engineering, Shenzhen Graduate School, Peking University
[4]School of Computer Science, Peking University
[5]Centre de Recherche Cerveau et Cognition (CERCO), UMR5549 CNRS–Université Toulouse 3
[6]University of Chinese Academy of Sciences

## Abstract

Spiking Neural Networks (SNNs) are distinguished from Artificial Neural Networks (ANNs) for their complex neuronal dynamics and sparse binary activations (spikes) inspired by the biological neural system. Traditional neuron models use iterative step-by-step dynamics, resulting in serial computation and slow training speed of SNNs. Recently, parallelizable spiking neuron models have been proposed to fully utilize the massive parallel computing ability of graphics processing units to accelerate the training of SNNs. However, existing parallelizable spiking neuron models involve dense floating operations and can only achieve high long-term dependencies learning ability with a large order at the cost of huge computational and memory costs. To solve the dilemma of performance and costs, we propose the mul-free channel-wise Parallel Spiking Neuron, which is hardware-friendly and suitable for SNNs' resource-restricted application scenarios. The proposed neuron imports the channel-wise convolution to enhance the learning ability, induces the sawtooth dilations to reduce the neuron order, and employs the bit-shift operation to avoid multiplications. The algorithm for the design and implementation of acceleration methods is discussed extensively. Our methods are validated in neuromorphic Spiking Heidelberg Digits voices, sequential CIFAR images, and neuromorphic DVS-Lip vision datasets, achieving superior performance over SOTA spiking neurons. Training speed results demonstrate the effectiveness of our acceleration methods, providing a practical reference for future research. Our code is available at Github.

## 1 Introduction

Inspired by the biological neural system, Spiking Neural Networks (SNNs) are regarded as the third-generation neural network models [1]. By emulating neuronal dynamics and spike-based communication characteristics in the brain [2], SNNs effectively capture temporal information and achieve event-driven efficient computation, providing a novel paradigm for building the spike-based machine intelligence [3].

Spiking neurons are the key component that distinguishes SNNs from Artificial Neural Networks (ANNs) [4]. They integrate input currents from synapses to membrane potentials by complex neuronal

---
*Corresponding author

39th Conference on Neural Information Processing Systems (NeurIPS 2025).

dynamics and fire spikes when the membrane potentials cross the threshold. These proceedings are generally described by several discrete-time difference equations [5, 6] in a formulation similar to the Recurrent Neural Networks. The discrete threshold-triggered firing mechanism induces the nondifferentiable problem and restricts the application of gradient descent methods. Recently, this problem has been resolved to a considerable degree by the surrogate gradient methods [7, 8, 9].

Deep SNNs [10, 11, 12] commonly use stateless synapses, i.e., the weights are shared across time-steps and outputs only depend on the inputs at the same time-step, to extract spatial features. Consequently, dynamic spiking neurons play a critical role in SNNs in extracting temporal features, and this has attracted many research interests. Most of the previous research focuses on increasing the neuron model complexity with learnable parameters [5, 13] or adaptive dynamics [14], which strengthens the model capacity but brings extra computation costs, making it unfriendly for resource-restricted neuromorphic hardware. Another issue is that traditional spiking neuron models run in a serial step-by-step mode, limiting the utilization of the powerful parallel computing capabilities of Graphics Processing Units (GPUs) and resulting in a slower training speed for SNNs compared to ANNs. Recently, parallelizable spiking neuron models [15, 16, 17, 18] have been proposed that overwhelm traditional serial models in running speed, showing a promising solution to accelerate the training of SNNs.

One of the most attractive characteristics of SNNs is that the multiply-accumulate (MAC) operations between binary spikes and synaptic weights can be superseded by accumulate (AC) operations during inference in neuromorphic chips [19]. However, in previous designs of spiking neurons, the computational costs of the neuronal dynamics have not been paid much attention. For instance, the Complementary Leaky Integrated-and-Fire (CLIF) neuron [14] introduces the computationally expensive sigmoid exponentiation, the Parallel Spiking Neuron (PSN), and the masked PSN [15] rely on the dense floating-point matrix multiplication. Compared to PSN and masked PSN, sliding PSN [15] only requires convolutional operations and demonstrates superior performance in handling time series with variable lengths. However, according to the study by [15], sliding PSN only achieves comparable performance to PSN when using a large convolutional kernel size, denoted as the order of the neuron, which significantly increases computational cost and memory usage. Moreover, these neurons still rely on massive multiply-accumulate (MAC) operations between floating-point neuronal weights and input currents.

In this article, we focus on designing a new variant of parallelizable spiking neuron models with hardware-friendly dynamics, low computation cost, and high long-term dependency learning ability. We propose an enhanced neuronal architecture named Multiplication-Free Channel-wise Parallel Spiking Neurons (mul-free channel-wise PSN), whose neuronal dynamics are shown in Figure 1, and validate its performance with state-of-the-art (SOTA) accuracy on temporal datasets. Our contributions are as follows.

1) To enhance the temporal information-capturing ability, we derive the sliding PSN by applying the channel-wise separable convolutions. To solve the dilemma of large temporal receptive fields and computational costs, we use dilated convolution. Compared to sliding PSN, our improvement does not introduce any additional floating point operations (FLOPs) and significantly reduces the inference memory.

2) To avoid the costly multiplication operations, we replace them with bit-shift operations, which are hardware-friendly for resource-restricted neuromorphic chips. The theoretical energy cost and area for hardware implementation with 8-bit integers (INT8) precision under 45nm CMOS is reduced $8\times$, from 0.2 $pJ$ and 282 $\mu m^2$ to 0.024 $pJ$ and 34 $\mu m^2$ [20], respectively.

3) We discuss the implementations of SNNs with mul-free channel-wise PSN on GPUs for efficient training. We propose an autoselect algorithm to choose the fastest implementations, which is practical for future research about accelerating parallelizable spiking neurons.

4) We achieve superior performance over other SOTA spiking neurons on various temporal tasks, validating the superior capability of the proposed methods in learning long-term dependencies.

## 2   Related Work

**Hardware-friendly Network Design.**   To deploy neural network models to edge devices such as mobile phones and Field Programmable Gate Arrays with limited energy, memory, and computational

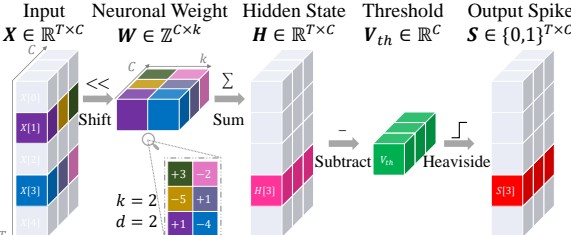

Figure 1: The neuronal dynamics of the mul-free channel-wise Parallel Spiking Neuron.

ability, a promising solution is hardware-friendly network design. Various methodologies have been proposed. Network quantization [21] quantizes the original weights and activations to low bits. Typical models in 8-bit integers require $4\times$ less memory consumption and achieve up to $4\times$ faster computation than models in 32-bit floating-point. Network pruning [22] prunes synapses and neurons to reduce the size of the model. Classic pruning methods can achieve $4\times$ compression ratio for ResNet-18 on ImageNet with about $5\%$ accuracy drop [23]. Knowledge distillation [24] employs a large teacher network to supervise the learning of a small student network, and the student network can obtain a competitive or even higher performance than the teacher network. Apart from the above universal methods, SNNs [1, 25] achieve extreme power efficiency by asynchronous event-driven computation in tailored neuromorphic chips. For instance, Intel Loihi [26] consumes $48\times$ energy efficiency than CPUs in solving the LASSO optimization problem; Tsinghua Tianjic [19] achieves up to $10^4$ times power efficiency over the Titan-Xp GPU when classifying the NMNIST dataset [27].

**Spiking Neuron Models.** The improvement of spiking neuron models provides a general method to upgrade SNNs, which attracts much interest in the research community. The Parametric Leaky Integrated-and-Fire (PLIF) spiking neuron [5] parameterizes the membrane time constant $\tau_m$ by a sigmoid function and can learn $\tau_m$ by gradient descent during training, showing better accuracy and lower latency than the traditional Leaky Integrated-and-Fire (LIF) neuron with fixed $\tau_m$. The Gated LIF (GLIF) neuron [13] assembles the learnable gate units to fuse different bio-features in the neuronal behaviors of membrane leakage, integration accumulation, and reset, achieving impressive performance by these rich neuronal patterns. The Complementary LIF (CLIF) neuron [28] introduces the complementary membrane potential into the LIF neuron, effectively capturing and maintaining information related to membrane potential decay. However, the sigmoid used in its neuronal dynamics cannot be removed during inference, which is costly for neuromorphic chips. The Parallel Spiking Neuron (PSN) family [15] and the Stochastic Parallelizable Spiking Neuron (Stochastic PSN) [16] are the first parallelizable spiking neuron models. These two models convert the iterative neuronal dynamics to a non-iterative formulation by removing the neuronal reset. Extending from PSN, several variants are proposed. The Parallel Multi-compartment Spiking Neuron (PMSN) [29] introduces multiple interacting substructures to enhance the learning ability over diverse timescales. The Parallel Spiking Unit (PSU) [30] adds a fully-connected layer with sigmoid activations inside the neuron to approximate the neuronal reset. These methods obtain performance gains over PSN in certain datasets, but increase the complexity of neuron models and slow down training speeds.

## 3 Preliminary

### 3.1 Traditional Spiking Neurons

In general, spiking neurons can be described by three discrete-time difference equations [5]:

$$H[t] = f(V[t-1], X[t]), \tag{1}$$

$$S[t] = \Theta\left(H[t] - V_{th}\right), \tag{2}$$

$$V[t] = \begin{cases} H[t] \cdot (1 - S[t]) + V_{reset} \cdot S[t], \text{hard reset} \\ H[t] - V_{th} \cdot S[t], \text{soft reset} \end{cases}. \tag{3}$$

Eq.(1) illustrates the neuronal charging process, where $X[t]$ is the input current at time-step $t$, extracted from the original spike input via a synapse layer (e.g., convolutional neural network or multi-layer perceptron). $H[t]$ and $V[t]$ are the membrane potentials before charging and after resetting

at time-step $t$, and $f$ is the charging equation specified for different spiking neurons. In Eq.(2), $\Theta(x)$ is the Heaviside step function, defined as $\Theta(x) = 1$ for $x \geq 0$ and $\Theta(x) = 0$ for $x < 0$. When $H[t]$ exceeds the threshold $V_{th}$, spiking neurons will fire spikes, and the membrane potential is reset as in Eq.(3). There are mainly two types of reset methods: hard reset will force the $V[t]$ to $V_{reset}$, while soft reset will subtract $V_{th}$ from $V[t]$.

## 3.2 Parallel Spiking Neuron

Fang et al. [15] found that for commonly used spiking neurons with a linear sub-threshold dynamic Eq.(1), such as the Integrate-and-Fire (IF) neuron and the LIF neuron, the neuronal dynamics could be expressed in a non-iterative form after removing the reset equation Eq.(3):

$$H[t] = \sum_{i=0}^{T-1} W[t][i] \cdot X[i], \tag{4}$$

where $W[t][i]$ is determined by Eq.(1). For example, $W[t][i] = \tau_m^{-1}(1 - \tau_m^{-1})^{t-i} \cdot \Theta(t-i)$ for the LIF neuron whose neuronal charging equation is:

$$H[t] = (1 - \tau_m^{-1}) \cdot V[t-1] + \tau_m^{-1} \cdot X[t]. \tag{5}$$

Fang et al. [15] extended Eq.(4) by setting $W[t][i]$ as a learnable parameter, and proposed the PSN with the following neuronal dynamics:

$$\boldsymbol{H} = \boldsymbol{W}\boldsymbol{X}, \qquad \boldsymbol{W} \in \mathbb{R}^{T \times T}, \boldsymbol{X} \in \mathbb{R}^T, \tag{6}$$

$$\boldsymbol{S} = \Theta(\boldsymbol{H} - \boldsymbol{V}_{th}), \qquad \boldsymbol{V}_{th} \in \mathbb{R}^T, \tag{7}$$

where $T$ is the sequence length. For simplicity, we ignore the batch dimension. The PSN does not involve iteration over time-steps. The core computation of the PSN is the matrix multiplication, which is highly optimized on GPUs and can be computed in parallel. Modified from the PSN, the sliding PSN is proposed by [15] with hidden states generating from the last $k$ inputs by a shared weight $\boldsymbol{W} \in \mathbb{R}^k$ across time-steps, whose neuronal dynamics are as follows:

$$H[t] = \sum_{i=0}^{k-1} W[i] \cdot X[t-k+1+i], \tag{8}$$

$$S[t] = \Theta(H[t] - V_{th}), \tag{9}$$

where $X[j] = 0$ for any $j < 0$ and $k$ is the order of the neuron. The sliding PSN can process input sequences with variable lengths, and the number of its parameters is decoupled with $T$. It can output $H[t]$ at the time-step $t$, while the PSN can only generate outputs after receiving the whole input sequence, making it more suitable for temporal tasks.

## 4 Methods

### 4.1 Channel-wise and Dilated Convolution

In PSN and sliding PSN, the weights of the neurons are shared across all channels. However, the visualization of feature maps from an SNN conducted by [5] implies that the difference between channels is huge, e.g., one channel extracts the edges and another channel extracts the background at all time-steps (refer to Figure S4 in [5] for more details). This coarse design concept of sharing weights across channels may fail to capture the subtle disparity of features in channels. To solve this issue, we extend the weights to channel-wise.

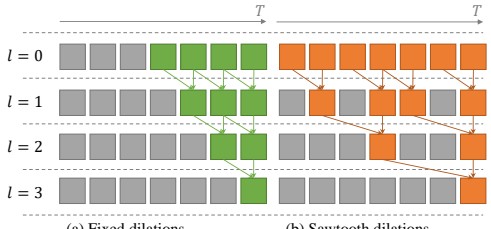

Figure 2: The temporal receptive field increases with depths at a slow rate in the sliding PSN with (a) fixed dilations and a fast rate in the channel-wise PSN with (b) sawtooth dilations.

To capture long-term dependency, the sliding PSN must use a large order $k$, resulting in a significant rise in computational cost and memory usage. We overcome this issue through the dilated convolution

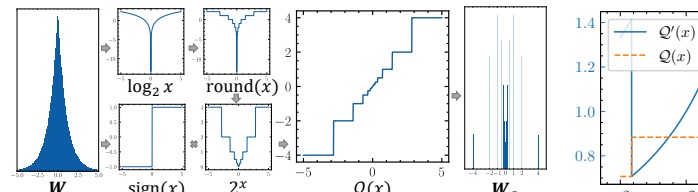

(a) The workflow of power-of-2 quantizer.

(b) Using Straight-Through Estimator (STE) to redefine $\mathrm{round}'(x)$ will cause a discrete gradient $\mathcal{Q}'(x)$.

Figure 3: Power-of-2 quantization and the gradient behavior of $\mathrm{round}'(x)$ under STE.

[31], where the convolution no longer processes consecutive inputs $(..., X[t-2], X[t-1], X[t])$, but instead $(..., X[t-2d], X[t-d], X[t])$, with $d > 1$ as the dilation rate.

Specifically, denote the input sequence as $\boldsymbol{X} \in \mathbb{R}^{T \times C}$, where $T$ is the sequence length and $C$ is the number of channels. We propose the channel-wise PSN with the following neuronal charging equation:

$$H[t][c] = \sum_{i=0}^{k-1} W[c][i] \cdot X[t - (k-1-i) \cdot d][c], \tag{10}$$

where $\boldsymbol{W} \in \mathbb{R}^{C \times k}$ is the learnable weight, $k$ is the order of the neuron and $d \in \mathbb{N}^+$ is the dilation rate. The channel-wise PSN has identical FLOPs to the sliding PSN, and the latter can be regarded as a simplified case with setting $W[0][i] = W[1][i] = ... = W[C-1][i]$ and $d = 1$ in Eq.(10).

Additionally, when using multiple layers of dilated convolutions, setting the same dilation rate will lead to the grid effect. An approach to solving this issue is to assign the dilation rate using a sawtooth wave-like heuristic [32]. Specifically, when constructing SNNs with channel-wise PSNs, we start by initializing with $d^0 = 1$ for the first spiking neuron layer, where the superscript represents the spiking neuron layer index. Then we update $d^l$ as:

$$d^{l+1} = d^l \mod 3 + 1. \tag{11}$$

This approach ensures that after stacking multiple layers, the convolution in the time domain could effectively incorporate inputs from all time-steps. Figure 2 shows how the temporal receptive field increases with depths with (a) fixed dilations $d = 1$ in the sliding PSN and (b) sawtooth dilations in the channel-wise PSN. The order is $k = 2$ in both cases. It can be found that, with increasing depths, both neurons achieve larger temporal receptive fields. However, the sliding PSN can only capture the last 4 time-steps with 3 layers, while the channel-wise PSN can capture the last 7 time-steps.

### 4.2 Multiplication-Free Neuronal Dynamics

To future reduce the internal computation costs of spiking neurons, we introduce the bit-shift operation to supersede multiplication, which has been successfully employed in quantized neural networks [33, 34]. In this way, the entire network can perform inference without any multiplication operations. For multiplication of IEEE 754-defined FP32/FP16 values $x$ with powers of 2 (denoted as $w$), the operation can be converted to a lower-bit integer addition to the floating-point exponent bits [35]. For integers $x$ multiplied by $w$, it can be achieved by the bit-shift operation, as shown in Eq.(12).

$$w \cdot x = x << \log_2(w), \tag{12}$$

where $<<$ is the left bit-shift operation. In particular, when $\log_2(w) < 0$, left shifting an negative number $\log_2(w)$ of bits is actually right shifting $>> |\log_2(w)|$. To employ the bit-shift operation, we quantize $\boldsymbol{W}$ in Eq.(10) to $\boldsymbol{W}_q$, whose elements are the power of 2, by an quantizing function $\mathcal{Q}$:

$$\boldsymbol{W}_q = \mathcal{Q}(\boldsymbol{W}) = \mathrm{sign}(\boldsymbol{W}) \cdot 2^{\mathrm{round}(\log_2(|\boldsymbol{W}|))}, \tag{13}$$

where $\mathrm{sign}(x)$ is the sign function and returns the sign (1 or $-1$) of the input $x$; $\mathrm{round}(x)$ is the rounding-to-nearest function. Figure 3a shows the workflow of Eq.(13).

Remarkably, the gradient of $\mathrm{round}(x)$ is zero almost everywhere, and other operations in Eq.(13) are differentiable. The standard practice is to employ the Straight-Through Estimator [36] to redefine its gradient as 1:

$$\mathrm{round}'(x) = 1. \tag{14}$$

Table 1: Parameters and computational costs of different spiking neurons during inference.

| Neuron | Params | Operations |
|--------|--------|------------|
| PSN | $T^2 + T$ | $(T^2 + T) \times \text{ADD}, T^2 \times \text{MUL}$ |
| Sliding PSN | $k + 1$ | $((T + \frac{1-k}{2}) \cdot k + T) \times \text{ADD}, ((T + \frac{1-k}{2}) \cdot k) \times \text{MUL}$ |
| Ours | $C \cdot (k+1)$ | $((T + \frac{1-k}{2}) \cdot k + T) \times \text{ADD}, ((T + \frac{1-k}{2}) \cdot k) \times \text{SHIFT}$ |

Then the gradient of Eq.(13) is:

$$\mathcal{Q}'(x) = \frac{1}{|x|} \cdot 2^{\text{round}(\log_2(|x|))}. \tag{15}$$

However, Eq.(15) is still unstable because $\text{round}(x)$ causes jump points and it oscillates around 0, shown in Figure 3b, which may cause the collapse. To avoid the numerical instability caused by Eq.(15), we redefine $\mathcal{Q}'(x)$ as a whole Straight-Through Estimator, rather than using Eq.(14) solely:

$$\frac{\partial \boldsymbol{W}_q}{\partial \boldsymbol{W}} = \mathcal{Q}'(\boldsymbol{W}) = \boldsymbol{1}. \tag{16}$$

The neuron model is called mul-free channel-wise PSN when using $\boldsymbol{W}_q$ in Eq.(13), and the complete neuronal dynamics is as follows, which is illustrated in Figure 1:

$$H[t][c] = \sum_{i=0}^{k-1} X[t - (k - 1 - i) \cdot d][c] << \log_2(W_q[c][i]), \tag{17}$$

$$S[t][c] = \Theta(H[t][c] - V_{th}[c]), \tag{18}$$

where $\boldsymbol{V}_{th} \in \mathbb{R}^C$ is the learnable channel-wise threshold, and $\log_2(\boldsymbol{W}_q) \in \mathbb{Z}^{C \times k}$ is quantized from $\boldsymbol{W}$ by Eq.(13). Note that $\log_2(\boldsymbol{W}_q)$ is solved beforehand and there are no $\log$ and multiplication operations during inference. Table 1 presents an analysis of the parameter and operation costs of various neurons, evaluated under $T$ time-steps, $k$ order and $C$ channels.

## 4.3 Training Acceleration

The motivation for proposing parallelizable spiking neuron models is to solve the slow training speed of SNNs on GPUs caused by the step-by-step iterations of traditional serial neuron models. Efficient implementation of mul-free channel-wise PSN, a typical 1-D convolution described in Eq.(17), requires elaborate consideration. To tackle this, we explore the implementations under two SNN data layouts: time-first and time-last. The time-first layout $(T, N, C, ...)$, widely adopted in SNN frameworks like SpikingJelly [6], accelerates stateless layers by fusing the $T$ (time-steps) and $N$ (batch size) dimensions, while $C$ denotes the channel dimension and "..." represents any additional dimensions. In comparison, the time-last layout $(N, C, ..., T)$ arranges $T$ as the last dimension.

A vanilla implementation of Eq.(17) leverages PyTorch's 1-D Convolution (*Conv1d*), which requires the shape of inputs as $(N, C, T)$. However, for both time-first and time-last layout, it is unavoidable to perform reshape operations $(T, N, C, ...) \rightleftharpoons (N*, C, T)$ and $(N, C, ..., T) \rightleftharpoons (N*, C, T)$ before and after the *Conv1d* to satisfy the shape requirement of *Conv1d*, where $N*$ represents the fusion of any additional dimensions in "..." into the $N$ dimension. Note that physical memory is 1-D, meaning that data in nonadjacent dimensions is also stored nonadjacent in physical memory. These reshape operations involve nonadjacent dimensions and require costly memory reading/writing (R/W) operations to create a new contiguous tensor in physical memory. Figure 4 illustrates examples of reshape operations with or without memory R/W.

To address the overhead caused by costly nonadjacent reshape operations in the neuron layer, we have developed several acceleration methods. For the time-first layout $(T, N, C, ...)$, we propose two methods. One is a custom CUDA kernel that directly performs convolutions along the $T$ dimension. Another leverages PyTorch's vectorizing map function *(Vmap)* to parallelize computations over the $C$ dimension, followed by matrix multiplication *(MM)* to process other dimensions. For the time-last layout $(N, C, ..., T)$, in addition to the custom CUDA kernel or *Vamp + MM* approach as in the time-first layout, we propose two additional methods. One uses the 2-D convolution to implement the 1-D convolution with the weight and stride as 1 to handle "..." dimension. Another applies *Vmap* to vectorize the $C$ dimension and employs *Conv1d* to handle other dimensions. Furthermore, we also

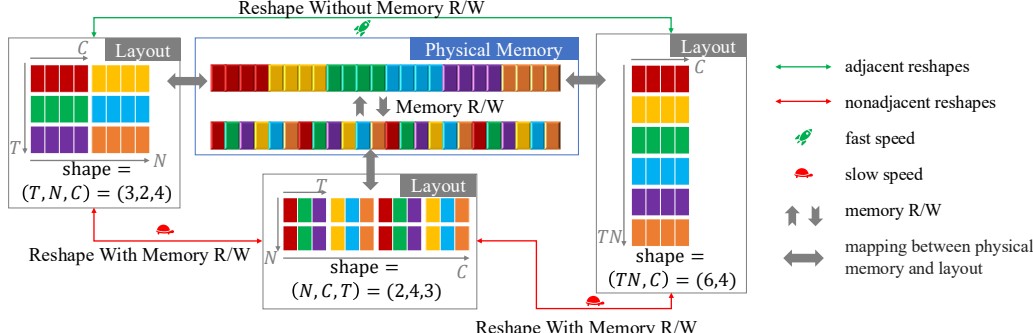

Figure 4: Reshape operations involving adjacent dimensions are free of memory reading/writing and are much faster than those involving nonadjacent dimensions.

elaborate on acceleration strategies for stateless layers in the time-last layout. Details for accelerating neuron and stateless layers are provided in Appendix A and B, respectively.

The choice of acceleration methods for neuron layers affects the data layout, which subsequently influences the performance of stateless layers. Moreover, the acceleration effect is dependent on input shapes and GPUs, making the process of selecting acceleration methods inherently empirical. To eliminate the need for manual selection, we design an autoselect acceleration algorithm, as shown in Algorithm 1. At the start of SNN training, the input sequence's shape is determined. Then the algorithm will run a benchmark over data layouts and acceleration methods, selecting the configuration with the highest execution speed. In Algorithm 1, each $t_{l,i}$ is evaluated through $2m+1$ repeated executions, with the average execution time of the last $m$ iterations serving as the measured runtime for the corresponding acceleration method. As the autoselect method is invoked only once during the entire training phase, its impact on overall train time is negligible.

---

**Algorithm 1** Autoselect acceleration algorithm

---

**Require:** An SNN stacked with $L$ layers $\{M_0, M_1, ..., M_{L-1}\}$. The layer $M_l$ has $n_l$ optional acceleration methods. The input sequence $\boldsymbol{X}_0$.

1: **for** $\Omega \leftarrow$ {time-first, time-last}
2:      Reshape $\boldsymbol{X}_0$ to $\Omega$
3:      $t_\Omega = 0$
4:      **for** $l \leftarrow 0, 1, ...L-1$
5:          **for** $i \leftarrow 0, 1, ..., n_l - 1$
6:              Record the current time $\mathcal{T}_0$
7:              Execute the forward propagation $\boldsymbol{Y}_l = M_l(\boldsymbol{X}_l)$
8:              Record the current time $\mathcal{T}_1$
9:              Randomize a tensor $\boldsymbol{Z}_l$ with the same shape as $\boldsymbol{Y}_l$
10:             Record the current time $\mathcal{T}_2$
11:             Execute the backward propagation $M'_l(\boldsymbol{Z}_l)$
12:             Record the current time $\mathcal{T}_3$
13:             $t_{l,i} = \mathcal{T}_1 - \mathcal{T}_0 + \mathcal{T}_3 - \mathcal{T}_2$
14:          Choose the faster method $a_{\Omega,l} = \text{argmin}_i(t_{l,i})$
15:          $t_\Omega \leftarrow t_\Omega + \min(t_{l,i})$

**Outputs:** The layout $\Omega^* = \text{argmin}_\Omega(t_\Omega)$ and the acceleration method $a_{\Omega*,l}$ for $M_l$

---

## 5 Experiments

In this section, we evaluate the mul-free channel-wise PSN on various kinds of datasets. We conduct the ablation experiments on the order $k$ and demonstrate that the sawtooth dilations can compensate for the long-term dependencies learning ability. Finally, we provide a training speed comparison to validate the efficiency of the autoselect algorithm.

### 5.1 Learning Long-Term Dependencies

We evaluate the long-term dependencies learning ability of mul-free channel-wise PSN in three widely used classification tasks, including the Spiking Heidelberg Digits (SHD) spoken digit dataset

Table 2: Comparison with the state-of-the-art SNN methods on the SHD dataset.

| Method | Network | Parallel | Accuracy(%) |
|---|---|---|---|
| Hammouamri et al. [39] | Two-layer FC + LIF + Learned Delay | ✗ | $95.07 \pm 0.24$ |
| Li et al. [30] | Four-layer FC + RPSU | ✓ | 92.49 |
| Chen et al. [29] | Two-layer FC + PMSN | ✓ | 95.10 |
| Yarga and Wood [16] | Two-layer FC + Stochastic PSN + Learned Delay | ✓ | 95.01 |
| **Ours** | Two-layer FC + Mul-free Channel-wise PSN + Learned Delay | ✓ | $\mathbf{95.50 \pm 0.36}$ |

Table 3: Comparison of test accuracy (%) of spiking neurons on sequential CIFAR datasets.

| Datasets | Ours | PMSN[29] | PSN[15] | Masked PSN[15] | Sliding PSN[15] | GLIF[13] | PLIF[5] | LIF |
|---|---|---|---|---|---|---|---|---|
| Sequential CIFAR10 | **91.17** | 90.97 | 88.45 | 85.81 | 86.70 | 83.66 | 83.49 | 81.50 |
| Sequential CIFAR100 | **66.21** | 66.08 | 62.21 | 60.69 | 62.11 | 58.92 | 57.55 | 53.33 |

Table 4: Comparison with the state-of-the-art ANN and SNN methods on the DVS-Lip dataset.

| Method | Frontend | Backend | Accuracy(%) |
|---|---|---|---|
| Tan et al. [38] | ResNet-18 (ANN) | BiGRU (ANN) | 72.1 |
| Bulzomi et al. [43] | Modified Spiking ResNet + PLIF | FC (Stateful Synapses) | 60.2 |
| Dampfhoffer et al. [42] | ResNet-18 (ANN) | BiGRU (ANN) | 75.1 |
| | Spiking ResNet-18 + PLIF | FC (Stateful Synapses) | 68.1 |
| | Spiking ResNet-18 + PLIF | SpikGRU2+ (Bi-direction + Sigmoid Gates + Ternary Spikes) | 75.3 |
| **Ours** | Modified Spiking ResNet-18 + Mul-free Channel-wise PSN | FC (Stateful Synapses) | 70.9 |

[37], the sequential CIFAR dataset, and the high spatial-temporal resolution automatic lip-reading DVS-Lip dataset [38]. These datasets cover the types of voices, images, and neuromorphic events.

Comparison between our method and previous SOTA SNN methods on the SHD dataset is shown in Table 2. Specifically, we replace the LIF neurons in the SNN-delay architecture [39] with our neurons. With sawtooth dilations and order $k = 2$, we achieve a test accuracy of $95.50 \pm 0.36\%$ under three seeds $(0, 1, 2)$.

Sequential image classification tasks have been commonly benchmarks to evaluate spiking neurons by [40, 15, 29]. In these tasks, images are fed into the model column by column, and the number of time-step is equal to the width of the images. We also conduct experiments on sequential CIFAR10 and CIFAR100 datasets. To ensure fairness, we fully employ the network architecture and hyperparameters as [15], only replacing the spiking neurons with ours. The results are shown in Table 3, where the data for PMSN is sourced from [29], maintaining the same architecture as well, while the data for other neurons is sourced from [15]. On the sequential CIFAR10 dataset, our mul-free channel-wise PSN outperforms PSN by $2.72\%$ and PMSN by $0.2\%$. Additionally, on the sequential CIFAR100 dataset, it surpasses PSN by $4\%$ and PMSN by $0.13\%$. Notably, the order of our neurons here we report is 16, while the order of sliding PSN and masked PSN is 32, $2\times$ than ours.

Furthermore, we demonstrate the capability of mul-free channel-wise PSN in processing complex neuromorphic DVS-Lip dataset, which comprises 100 classes and consists of 19871 samples, each containing approximately $10^4$ events with a spatial resolution of $128 \times 128$. Half of the words in the dataset are visually similar pairs in the LRW dataset [41] (e.g., "America" and "American"). The training and testing sets are derived from different speakers, posing a challenge for the model to exhibit robust generalization capabilities with respect to speaker characteristics.

Currently, the SOTA accuracy of 75.3% on the DVS-Lip dataset is achieved by [42] using a Spiking ResNet-18 with the channel-wise PLIF neurons fronted, a SpikGRU2+ backend, and events are integrated into 90 frames ($T = 90$). In our experiments, we introduce several modifications to the frontend. We replace the PLIF neurons with our neurons and remove spiking neurons from the pooling layers, referring to this architecture as Modified Spiking ResNet-18. As Table 4 shows, our method achieves 70.9% accuracy and is only second to [42] with SpikGRU2+ backend. It is worth noting that SpikGRU2+ is bi-directional with two groups of separate hidden states, employs sigmoid gates with floating activations, and outputs ternary spikes $(-1, 1, 0)$, which is not a pure SNN module and might be difficult to deploy to neuromorphic chips. The accuracy we report here is based on the neuron order $k = 2$ and sawtooth dilations, indicating that our method can effectively capture rich historical information with a small order even when handling tasks involving long-time sequences.

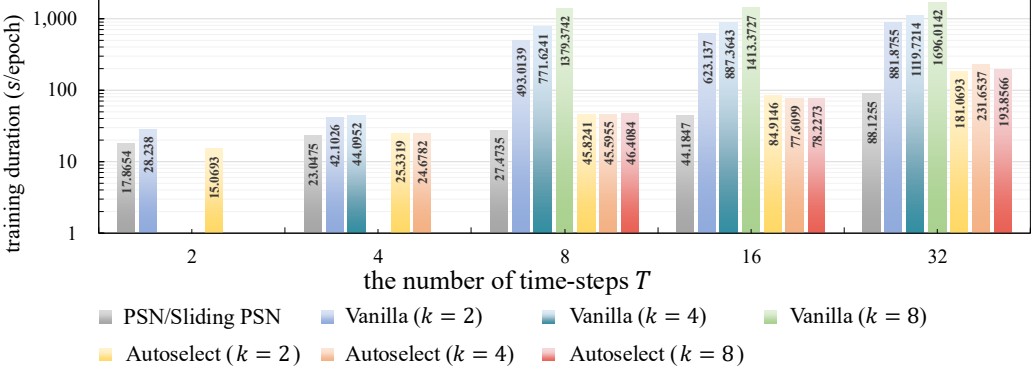

Figure 6: Comparison of training speed on CIFAR100.

## 5.2 Ablation Study

To validate that our neurons can effectively approximate the effect of a larger receptive field with a smaller order $k$ through sawtooth dilations, we conduct ablation experiments on the sequential CIFAR100 and pixel CIFAR10 classification tasks.

Figure 5 (a) illustrates the accuracy curves of mul-free channel-wise PSN and other neurons on the sequential CIFAR100 dataset, with the highest accuracy marked by a red ★. When the order is 2, the accu-

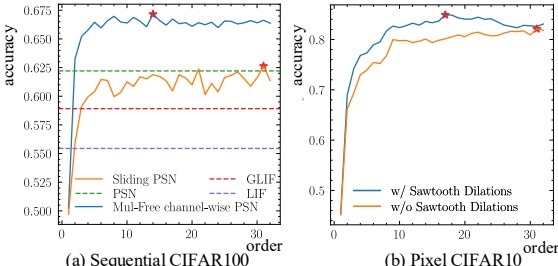

Figure 5: The order-accuracy curves on (a) the sequential CIFAR100 and (b) the pixel CIFAR10.

racy of our neuron already significantly surpasses the whole PSN family. Furthermore, when the order increases to 3 or more, the accuracy remains roughly around 66%. It is evident that our neuron is more robust than the sliding PSN, as it does not exhibit the issue of fluctuating accuracy while increasing order.

To evaluate the effectiveness of sawtooth dilations, we conduct an ablation study on the pixel CIFAR10 classification task. In this task, images are flattened into one-dimensional vectors as time series inputs to the network. Thus, the number of time-step is $T = 1024$. We adopt the same network structure as [29]. Figure 5 (b) illustrates the accuracy curves with/without sawtooth dilations. It can be observed that the accuracy with sawtooth dilations is consistently higher than that without. Additionally, when the order $k$ is small, which is a practical case for deployment, the accuracy of our neuron with sawtooth dilations is much higher. These results validate that the sawtooth dilations compensate for the effect of large receptive fields when using a small $k$.

## 5.3 Training Acceleration

We compare the training speed of PSN and the mul-free channel-wise PSN implemented by the autoselect Algorithm 1. The vanilla implementation, using reshape and *Conv1d* operations for neuron layers and time batch dimension fusion for stateless layers, is also compared. The experiments are carried out on a Debian GNU/Linux 11(bullseye) server with an Intel(R) Xeon(R) Platinum 8336C CPU, an Nvidia A100-SXM4-80GB GPU and 32GB RAM. Following the original PSN [15] settings, we use the batch size of 128 on the CIFAR dataset.

The training duration (*s*/epoch) of different neurons under different order $k$ on CIFAR100 is shown in Figure 6. Note that both PSN and sliding PSN are implemented by matrix multiplication in GPUs [15], their speeds are identical and decoupled with $k$. The results show that our autoselect algorithm greatly improves the efficiency of mul-free channel-wise PSN and achieves a much faster training speed than the vanilla implementation. When $T \leq 4$, our method is comparable to PSN/sliding PSN. In this case, the matrix is nearly stripped in PSN/sliding PSN because $T$ is much less than other dimensions, causing inefficient matrix multiplications. When $T$ continuously increases, our method is slower, which is caused by the fact that the quantization induces additional overhead, and memory R/W caused by *Vmap* operations for processing inputs/outputs in our SNNs is slower than

Table 5: Step-by-step inference memory on the sequential CIFAR100 ($T = 32$) dataset.

| Neuron | $k$ | Accuracy(%) | Memory(MB) |
|---|---|---|---|
| Sliding PSN | 32 | 62.11 | 2635 |
| **Ours** | 4 | 65.77 | 547 |

Table 6: Computational energy comparison on a single CIFAR100 Image.

| Neuron Layer | | | Synaptic Layer | | Total Energy |
|---|---|---|---|---|---|
| Neuron | Operations | Energy ($\mu$J) | Operations | Energy ($\mu$J) | ($\mu$J) |
| PSN | $1.91 \times 10^7$ MUL $1.97 \times 10^7$ ADD | 88.56 | $0.041 \times 10^6$ FLOPs $3.194 \times 10^6$ SOPs | 3.06 | 91.62 |
| **Ours** | $7.32 \times 10^6$ SHIFT $7.92 \times 10^6$ ADD | 8.08 | $0.041 \times 10^6$ FLOPS $2.660 \times 10^6$ SOPs | 2.58 | 10.66 |

the dimension fusion. Nonetheless, the speed gaps are not significant. The autoselect algorithm itself introduces only negligible time overhead, contributing $1.17\%, 0.39\%, 0.27\%$ for $T = 2, 8, 32$, respectively. Further results for different batch sizes and GPUs can be found in Appendix G.

## 5.4 Inference Memory Analysis

Both our approach and sliding PSN require storing an input sequence whose length is proportional to the neuron order $k$. According to [15], sliding PSN typically needs a large $k$ to maintain stable performance on long-term dependencies. In contrast, our method can achieve more stable and better performance with a much smaller $k$, as shown in Fig. 5. By substantially reducing the required neuron order, our approach shortens the length of the stored input sequence and significantly lowers memory consumption during inference. As demonstrated in Table 5, this leads to a substantial reduction in memory overhead for storing input sequences, making our solution more suitable for deployment on resource-constrained devices.

## 5.5 Inference Energy Estimation

Assume implemented on the 45 nm CMOS technology, where a 32-bit floating-point (FP32) multiplication (MUL) and addition (ADD) operation consumes 3.7 pJ and 0.9 pJ, respectively [44]. In comparison, for the shift operation of a 32-bit fixed-point (FIX32) and a power-of-two number, it requires only 0.13 pJ [44]. For the multiplication of a FP32 number with a power-of-two number, it can be performed by one single lower-bit integer addition, which is also energy efficient [35]. Here we use the FIX32 shift energy as an approximation. Based on the sequential CIFAR100 models (see Appendix D), we estimate the average computational energy of PSN and our method for processing a single CIFAR100 image, as shown in Table 6. Our method achieves nearly $9\times$ lower energy consumption compared to PSN, demonstrating a significant advantage in hardware efficiency. Notably, our energy estimation considers the cost of neuronal layers, whereas most previous studies only account for synaptic layers. Moreover, we observe that for PSN, the energy consumption of the neuronal layer is significantly higher than that of the synaptic layer, which is mainly due to the dense floating-point matrix multiplications involved. Details of the energy estimation procedure can be found in Appendix F.

## 6 Conclusion

In this paper, we introduce a novel parallelizable spiking neuron model named mul-free channel-wise PSN, which employs the channel-wise convolutions to process the input sequences, avoids the large neuron order by sawtooth dilations, and gets rid of floating multiplications by efficient bit-shift operations. The considerations of accelerating the training of SNNs with the proposed neuron models are also discussed in detail. Experimental results demonstrate that mul-free channel-wise PSN achieves significant performance improvements in temporal classification tasks, showcasing its superior capability to capture long-term dependencies. Our methods solve the dilemma of performance and computational costs of spiking neuron models, and our acceleration methods will benefit future research as a practical reference.

## Acknowledgments and Disclosure of Funding

This work is supported by National Science and Technology Innovation 2030 Major Project (No. 2025ZD0215501), Guangdong S&T Programme 2024B0101010003 and National Natural Science Foundation of China (62236009).

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

# Appendix

## A Acceleration Details of Spiking Neuron Layer

### A.1 Acceleration Methods

Based on the aforementioned background, we design the following accelerating implementations as candidates:

**(1) *Time-last + Vmap + Conv1d*:** this method uses the vectorizing map function (*Vmap*) in PyTorch to vectorize *Conv1d* to process the input sequence with the $(N, C, *, T)$ layout over the "$*$" dimension. The reshape operations $(N, C, ..., T) \rightleftharpoons (N, C, *, T)$ are nearly cost-free because the reshaped dimensions are adjacent physically.

**(2) *Time-last + Conv2d*:** this method is similar to the implementation (1), but processes the input sequence with the $(N, C, *, T)$ by *Conv2d* and sets the weight and stride in the "$*$" dimension as 1, rather than by *Vmap*.

**(3) *Time-first/last + Vmap + MM*:** this method uses *Vmap* to vectorize matrix multiplication (*MM*) to process inputs over channels ($c$ in Eq.(17)). Refer to Appendix A.2 for more details about how the weights for *MM* are generated.

**(4) *Time-first/last + Custom CUDA Kernel*:** this method avoids reshape operations and can be used for any memory layout. However, the convolutions in PyTorch are highly optimized, i.e., implemented by the official NVIDIA CUDA Deep Neural Network (cuDNN) library, which might be much faster than custom implementations. Refer to Appendix A.3 for more details.

Note that if the spiking neuron layer implementations adopt the time-last layout, the stateless layers should also use the same layout. Otherwise, reshape operations between time-first and time-last layouts will cause great latency. Correspondingly, the time batch dimension fusion method to accelerate stateless layers in SpikingJelly cannot be applied. In Appendix B, we fully discuss the acceleration method of the stateless layer for the time-last layout.

### A.2 Details of Time-first/last + Vmap + MM

In Eq.(17), weight of the standard 1D convolution $\boldsymbol{W}_q$ is shaped as $[C, k]$. The standard 1D convolution operation could be implemented by matrix multiplication and vectorizing map. When the input sequence $\boldsymbol{X} \in \mathbb{R}^{T \times N}$ arrives, where the sequence length $T$ is known, for the time-first data layout, the weight matrix $\boldsymbol{A} \in \mathbb{R}^{C \times T \times T}$ can be generated as:

$$\boldsymbol{A}[:][i][j] = \begin{cases} W_q[:][k - 1 - \frac{i-j}{d}], & i - d(k-1) \leq j \leq i \ \& \ (i-j)\%d = 0 \\ 0, \text{otherwise} \end{cases}, \qquad (19)$$

where $[:]$ means the slice operation.

Similarly, for the time-last data layout, the weight matrix $A \in \mathbb{R}^{C \times T \times T}$ can be generated as:

$$\boldsymbol{A}[:][j][i] = \begin{cases} W_q[:][k - 1 - \frac{i-j}{d}], & i - d(k-1) \leq j \leq i \ \& \ (i-j)\%d = 0 \\ 0, \text{otherwise} \end{cases}. \qquad (20)$$

Applying the vectorizing map to the input sequence and the weight matrix across the channel dimension, the membrane potential $\boldsymbol{H}$ can be calculated through the matrix multiplication operation over channels in parallel:

$$\boldsymbol{H}[c] = \begin{cases} \boldsymbol{A}[c]\boldsymbol{X}[c], \text{time-first layout} \\ \boldsymbol{X}[c]\boldsymbol{A}[c], \text{time-last layout} \end{cases}. \qquad (21)$$

### A.3 Details of Time-first/last + Custom CUDA Kernel

Suppose $\boldsymbol{X}$ is the input sequence, $\boldsymbol{H}$ is the hidden states, and $\boldsymbol{\delta}^{\boldsymbol{H}}$ is the gradient with respect to $\boldsymbol{H}$, obtained by automatic differentiation in PyTorch, all of them are shaped as $(T, N, C, H, W)$ or

$(N, H, W, C, T)$. Suppose $\boldsymbol{W}$ and $\boldsymbol{b}$ are the weight and bias of the convolution, shaped as $[C, k]$ and $[C]$, respectively.

The process of forward propagation can be represented as:

$$\boldsymbol{H} = \text{pad}(\boldsymbol{X}, (k-1, 0)) \star \boldsymbol{W} + b, \tag{22}$$

where pad represents padding $k - 1$ zeros on the left of $\boldsymbol{X}$ over the time dimension $T$, and $\star$ denotes the convolution operation on the $T$ dimension of $X$ using $W$.

The process of backward propagation can be represented as:

$$\frac{\partial L}{\partial \boldsymbol{X}} = \text{pad}(\boldsymbol{\delta^H}, (0, k-1)) \star \text{flip}(\boldsymbol{W}), \tag{23}$$

$$\frac{\partial L}{\partial \boldsymbol{W}} = \text{pad}(\boldsymbol{X}, (k-1, 0)) \star \boldsymbol{\delta^H}, \tag{24}$$

$$\frac{\partial L}{\partial b} = \sum_{t,n,h,w} \boldsymbol{\delta^H_{t,n,n,w}}, \tag{25}$$

where flip($\boldsymbol{W}$) represents flip $\boldsymbol{W}$ left and right along the $k$ dimension.

Beyond PyTorch (cuDNN), a custom CUDA implementation for two data layouts is also considered. To avoid the reshape and incident memory copying, we design CUDA kernels by OpenAI Triton for processing both data layouts directly. Specifically, we manually implement the Eqs.(22)-(25) using the Triton framework. We design a custom autograd function, where the aforementioned kernel functions are called in the forward and backward methods. Note that the convolution operation in Eq.(23) is consistent with that in Eq.(22), so the triton kernel function remains the same. Taking the time-first layout as an example, Eq.(22) and (24) could be implemented as Algorithms 2 and 3, respectively.

---

**Algorithm 2** Triton forward kernel
___
**Input:** The input sequence pointer $X_{ptr}$, weight matrix pointer $W_{ptr}$, the output sequence pointer $H_{ptr}$, point to the first address of tensors, shaped ad $[T + k - 1, N, C, H, W]$, $[C, k]$ and $[T, N, C, H, W]$, respectively.
1: Utilize the triton autotune method to determine the BLOCK_SIZE_NHW($BN$) and BLOCK_SIZE_C($BC$)
2: Calculate the offset $\boldsymbol{X}_{offset}$, $\boldsymbol{W}_{offset}$ and $\boldsymbol{H}_{offset}$ of each thread, shaped as $[BN, BC, T, k]$, $[1, BC, 1, k]$ and $[BN, BC, T]$, respectively
3: Load values of $X_{ptr} + \boldsymbol{X}_{offset}$ and $W_{ptr} + \boldsymbol{W}_{offset}$ from memory to SRAM tensors $\boldsymbol{X}$ and $\boldsymbol{W}$
4: Utilizing the broadcasting mechanism, perform the element-wise multiplication of $\boldsymbol{X}$ and $\boldsymbol{W}$
5: Sum the output $\boldsymbol{H}$ along the $k$ dimension
6: Store the values of $\boldsymbol{H}$ to $H_{ptr} + \boldsymbol{H}_{offset}$ address

---

**Algorithm 3** Triton grad of weight kernel
___
**Input:** The grad of output pointer $O_{ptr}$, the input sequence pointer $X_{ptr}$, the grad of weight pointer $W_{ptr}$, point to the first address of tensors, shaped ad $[T, N, C, H, W]$, $[T + k - 1, N, C, H, W]$ and $[C, k]$, respectively.
1: Utilize the triton autotune method to determine the BLOCK_SIZE_NHW($BN$) and BLOCK_SIZE_C($BC$)
2: Calculate the offset $\boldsymbol{O}_{offset}$, $\boldsymbol{X}_{offset}$ and $\boldsymbol{W}_{offset}$ of each thread, shaped as $[BN, BC, T, 1]$, $[BN, BC, T, k]$ and $[BC, k]$, respectively
3: Load values of $O_{ptr} + \boldsymbol{O}_{offset}$ and $X_{ptr} + \boldsymbol{X}_{offset}$ from memory to SRAM tensors $\boldsymbol{O}$ and $\boldsymbol{X}$
4: Utilizing the broadcasting mechanism, perform the element-wise multiplication of $\boldsymbol{O}$ and $\boldsymbol{X}$
5: Sum the grad of weight $\boldsymbol{W}$ along the $T$ and $k$ dimensions
6: Atomic add the values of $\boldsymbol{W}$ to $W_{ptr} + \boldsymbol{W}_{offset}$ address

---

# B   Acceleration Details of Stateless Layer

Stateless layers include the convolutional, batch normalization, pooling, and linear layers. When using the time-first data layout, the stateless layers can be accelerated by fusing the time dimension and the

batch dimension in SpikingJelly. More specifically, the data layout changes as $(T, N, *) \rightleftharpoons (TN, *)$ before and after processing of the stateless layers. Then GPUs regard the time-step dimension as the batch dimension, leading to fully parallel computing over time-steps. It is worth noting that the dimension fusion is nearly no cost because the time and batch dimensions are physically adjacent in memory. The reshape operation only changes the view of tensors and does not involve memory copying.

When using the time-last layout, the dimension fusion method of SpikingJelly cannot be applied except for the batch normalization layer, which only requires that the channel dimension be the 1-th dimension. Both layouts can be satisfied by a reshape without additional memory R/W. For the convolutional and pooling layer, we introduce two new methods: the vectorizing map provided in PyTorch and the high-dimension convolution/pooling that has been used in the Lava framework, a software framework for neuromorphic computing.

The vectorizing map vectorizes the stateless layers to process the input sequence with the $(N, ..., T)$ layout over the last dimension $T$, then the computation over time-steps is in parallel. This method actually implies a reshape operation $(N, ..., T) \rightleftharpoons (T, N, ...)$ when splitting and concatenating the sequence. The high-dimension convolution/pooling uses the $(n + 1)-$D convolution/pooling to implement the $n$-D convolution/pooling with a weight of 1 and a stride of 1 in the additional dimension. This method is also in parallel, while the main drawback is that the high-dimension convolution/pooling is complex and not as efficient as the dimension fusion method [6].

## C Neuron Quantization

To reduce the internal covariate shift along the temporal and batch dimensions, and increase the numerical stability of the model, we use batch normalization to implement the learnable threshold $V_{th}$. Eq.(18) could be rewrite as:

$$S[t][c] = \Theta \left( \gamma[c] \frac{H[t][c] - \mu_{\mathcal{B}}[c]}{\sqrt{\sigma_{\mathcal{B}}^2[c] + \epsilon}} + \beta[c] \right), \tag{26}$$

where $\gamma \in \mathbb{R}^C$ and $\beta \in \mathbb{R}^C$ are the learnable weight, initialized as $1$ and $-1$. $\mu_{\mathcal{B}} \in \mathbb{R}^C$ and $\sigma_{\mathcal{B}}^2 \in \mathbb{R}^C$ are the mean and variance of the input over the dimension $C$. Specifically, at train time, they are the mean and biased variance of the input sequence; at inference time, they are the moving average of the mean and unbiased variance of the input sequence on the training stage, which means $\mu_{\mathcal{B}}$ and $\sigma_{\mathcal{B}}^2$ are invariant during inference.

Since our quantization goal is to use the efficient bit-shift operation to replace the multiplication, the convolution layer and the batch normalization layer could be fused to reduce computation during inference, so we need to quantize the fused weights during training. The formula for the fusion of convolution and batch normalization can be represented as follows:

$$\boldsymbol{W}_f = \frac{\boldsymbol{\gamma}}{\sqrt{\boldsymbol{\sigma}_{\mathcal{B}}^2 + \epsilon}} \cdot \boldsymbol{W}, \tag{27}$$

$$\boldsymbol{b}_f = \boldsymbol{\beta} - \frac{\boldsymbol{\gamma} \cdot \boldsymbol{\mu}_{\mathcal{B}}}{\sqrt{\boldsymbol{\sigma}_{\mathcal{B}}^2 + \epsilon}}. \tag{28}$$

Thus, $\boldsymbol{W}$ in Eq.(13) is actually $\boldsymbol{W}_f$ in Eq.(27). To implement the quantization of fused weight, during the training stage, input sequence $\boldsymbol{X}$ is first passed to the convolutional layer, resulting in the intermediate variable $\boldsymbol{T}$ to update the $\boldsymbol{\mu}_{\mathcal{B}}$ and $\boldsymbol{\sigma}_{\mathcal{B}}^2$ in Eq.(27) and Eq.(28). Then, we use $\boldsymbol{W}_f$ as $\boldsymbol{W}$ in Eq.(13) and $\boldsymbol{b}_f$ as $\boldsymbol{V}_{th}$ in Eq.(18), perform the convolution operation on the input $\boldsymbol{X}$ twice. After the training is completed, $\boldsymbol{\mu}_{\mathcal{B}}$ and $\boldsymbol{\sigma}_{\mathcal{B}}^2$ is fixed, so we could directly use the quantized $\boldsymbol{W}_f$ and $\boldsymbol{b}_f$ as the weight and bias of the convolution layer, performing the convolution operation only once.

## D Network Structure

Table 7 illustrates the details of the network structure for different datasets. c128k3s1 represents convolution layer with output channels = 128, kernel size = 3 and stride = 1, BN is the batch normalization. SN is the mul-free channel-wise PSN, APk2s2 is the avg-pooling layer with kernel

size $= 2$ and stride $= 2$, FC256 represents the fully connected layer with output feature $= 256$. RB128 is the residual block with output channels $= 128$, Dcls256 is the dilated convolution with learnable spacings with output channels $= 256$, DP is the dropout layer. LIF(Vth=1e9) represents a LIF spiking neuron the threshold $= 1e9$, and the membrane potential is the output, which could be thought of the moving average of the input current. 3D_c64k577s122p233 represents the 3D convolution layer with output channels $= 64$, kernel size $= (5, 7, 7)$, stride $= (1, 2, 2)$ and padding $= (2, 3, 3)$. AAPk1 is the adaptive avg-pooling layer with output feature $= 1$. Stateful FC 100 is an FC layer with stateful synapses.

Table 7: Network structure for different datasets.

| Dataset | Network structure |
|---|---|
| Sequential CIFAR10/CIFAR100 | {{c128k3s1-BN-SN}*3-APk2s2}*2-FC256-SN-FC10/100 |
| Pixel CIFAR10 | FC128-BN-SN-{RB128}*2-APk4s4-FC256-BN-SN-{RB256}*2-FC10 |
| SHD | {Dcls256-BN-SN-DP}*2-Dcls20-LIF(Vth=1e9) |
| DVS-Lip | 3D_c64k577s122p233-APk3s2p1-{RB64}*2-{RB128}*2-{RB256}*2-{RB512}*2-AAPk1-DP-Stateful FC 100 |

# E    Setting of Experiments

The main hyper-parameters for different datasets are shown in Table 8. Other training options are listed as follows.

**Sequential CIFAR** The data augmentation techniques include random mixup with $p = 1$ and $\alpha = 0.2$, random cutmix with $p = 1$ and $\alpha = 1$, random choice between the two mix methods with $p = 0.5$, random horizontal flip with $p = 0.5$, trivial augmentation, normalization, random erasing with $p = 0.1$, and label smoothing with the amount 0.1 [15]. The number of channels is 128. The surrogate function is the arctan surrogate function $\sigma(x) = \frac{\alpha}{2(1+(\frac{\pi}{2}\alpha x)^2)}$ with $\alpha = 2$.

**Pixel CIFAR** All parameters and experimental settings are the same as Sequential CIFAR.

**SHD** Augmentation methods include spatio jitter with var 0.55, uniform noise with number $n = 35$, drop event with $p = 0.05$, drop event chunk with $p = 0.3$ and max drop chunk length $l = 0.02$. The surrogate function is also the arctan surrogate function with $\alpha = 5$.

**DVS-Lip** The data augmentation techniques include center cropped size $= 96 \times 96$, then random cropped size $= 88 \times 88$, random horizontal flip with $p = 0.5$, 2D spatial mask with mask num $= 4$ and maximum length$= 20$, random choice between zoom in and zoom out with $p = 0.5$ and max scale $= 26$, temporal mask with mask num $= 6$ and maximum length $= 18$ [42]. The surrogate function is $\sigma(x) = \frac{1}{1+\alpha x^2}$ with $\alpha = 10$.

Table 8: Training hyper-parameters for different datasets.

| Dataset | Optimizer | Batch Size | Epoch | Learning Rate | Scheduler |
|---|---|---|---|---|---|
| Sequential CIFAR10/100 | AdamW | 128 | 256 | 0.001 | CosineAnnealingLR |
| Pixel CIFAR10 | AdamW | 128 | 128 | 0.001 | CosineAnnealingLR |
| SHD | Adam (wd=1e-5) | 128 | 150 | 5e-4 for weights 0.05 for delay | CosineAnnealingLR for weights OneCycleLR for delay |
| DVS-Lip | Adam (wd=1e-4) | 32 | 200 | fixed 3e-4 for 0-100 epochs (1e-4, 5e-6) for 100-200 epochs | CosineAnnealingLR |

# F    Details of the Energy Estimation

By combining the neuron operation counts in Table 1 with the corresponding per-operation energy consumption, the theoretical neuronal energy consumption can be estimated, as shown in Table 9.

Table 9: Operations and energy of different spiking neurons during inference.

| Neuron | Operations | Energy (pJ) |
|--------|-----------|-------------|
| PSN | $(T^2 + T) \times$ ADD, $T^2 \times$ MUL | $4.6T^2 + 0.9T$ |
| Ours | $((T + \frac{1-k}{2}) \cdot k + T) \times$ ADD, $((T + \frac{1-k}{2}) \cdot k) \times$ SHIFT | $1.03(T + \frac{1-k}{2}) \cdot k + 0.9T$ |

When the neuron order $k$ reaches its maximum value $T$, the maximum theoretical energy consumption of our neuron is $0.515T^2 + 1.415T$, which is approximately $9\times$ lower than that of PSN for large $T$. This result clearly demonstrates the superior energy efficiency of our approach in hardware-constrained scenarios.

Further, we use the sequential CIFAR100 Network structure (see Appendix D for more details) to evaluate the energy consumption on a single CIFAR100 Image. Following [11], we use Eq.(29) to calculate the energy of synaptic layers, where $FL^1_{SNNConv}$ is the FLOPs of the first layer to encode static RGB images into spike-form, $N$ is the number of convolutional layers, and $M$ is the number of fully connected layers. $fr$ in Eq.(30) is the firing rate of the input spike sequence of every synaptic layer, and here is the average value of the trained network across the entire test dataset. Assume the MAC and AC operations are implemented on the 45nm hardware, $E_{MAC} = 4.6pJ$ and $E_{AC} = 0.9pJ$. Following the experimental setup specified in Table 3, the neuron order $k$ of our neuron is 16. Detailed energy analysis is shown in Table 6, where the FLOPs and SOPs of the synaptic layers refer to the $FL^1_{\text{SNN Conv}}$ and $(\sum_{n=2}^{N} \text{SOP}^n_{\text{SNN Conv}} + \sum_{m=1}^{M} \text{SOP}^m_{\text{SNN FC}})$ in the Eq.(29), respectively.

$$E_{\text{Synaptic}} = E_{MAC} \times \text{FL}^1_{\text{SNN Conv}} + E_{AC} \times \left( \sum_{n=2}^{N} \text{SOP}^n_{\text{SNN Conv}} + \sum_{m=1}^{M} \text{SOP}^m_{\text{SNN FC}} \right) \quad (29)$$

$$\text{SOPs}(l) = fr \times T \times \text{FLOPs}(l) \quad (30)$$

## G   Experimental Results

Table 10 presents the original data depicted in Figure 5, illustrating the performance of mul-free channel-wise PSN across varying orders and datasets.

In addition, we report extra results on the H20 GPU and the A100 GPU with different batch sizes in Tab.11. The results and conclusions are consistent with using the A100 GPU and batch size 128. Notably, the training duration of our neuron is with the quantization operation, i.e., perform the convolution operation twice, so it is inherently slower than PSN on the training stage.

For the training speed across different neuron implementations (PSN, vanilla, and autoselect), each method is trained for two epochs, and the training durations of the second epoch are set as the results. In this way, the results have accounted for statistical validity and warm-up of GPUs and eliminated the influence of model loading and other initialization processes. Hence, the reported speed is *the speed at the steady state*.

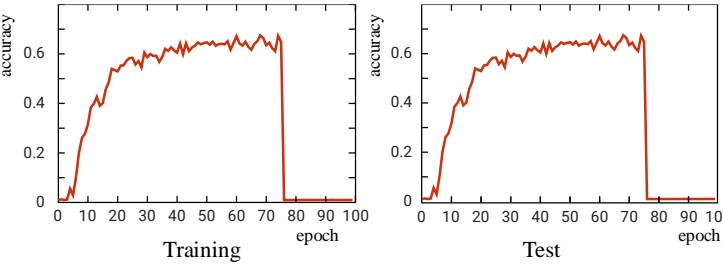

Figure 7: The training and testing accuracy curves with gradient of Eq.(14) on the DVS-Lip Dataset.

Table 10: Test accuracy (%) of the multiplication-free channel-wise PSN, corresponding to the data presented in Figure 5.

| Dataset / Order | Sequential CIFAR100 | Pixel CIFAR 10 (w/o dilation) | Pixel CIFAR 10 (w/ dilation) |
|---|---|---|---|
| 1 | 50.24 | 45.15 | 45.33 |
| 2 | 63.25 | 66.21 | 68.96 |
| 3 | 65.21 | 69.33 | 74.21 |
| 4 | 65.77 | 72.97 | 76.82 |
| 5 | 66.45 | 73.90 | 77.31 |
| 6 | 65.96 | 75.44 | 78.92 |
| 7 | 66.58 | 75.21 | 79.49 |
| 8 | 66.97 | 76.72 | 81.71 |
| 9 | 66.48 | 79.97 | 82.67 |
| 10 | 66.38 | 79.75 | 82.36 |
| 11 | 66.87 | 79.80 | 82.74 |
| 12 | 66.53 | 79.39 | 83.24 |
| 13 | 66.06 | 79.65 | 82.85 |
| 14 | 67.15 | 80.07 | 83.58 |
| 15 | 66.60 | 79.37 | 83.32 |
| 16 | 66.21 | 79.83 | 83.65 |
| 17 | 66.73 | 80.13 | 84.89 |
| 18 | 66.32 | 80.51 | 84.84 |
| 19 | 66.41 | 80.89 | 84.28 |
| 20 | 66.10 | 80.56 | 84.05 |
| 21 | 66.41 | 81.19 | 84.04 |
| 22 | 66.23 | 81.44 | 84.43 |
| 23 | 66.45 | 81.00 | 84.01 |
| 24 | 65.98 | 80.71 | 84.11 |
| 25 | 66.56 | 80.78 | 83.49 |
| 26 | 66.53 | 81.26 | 82.85 |
| 27 | 66.30 | 81.61 | 82.60 |
| 28 | 66.40 | 81.58 | 82.33 |
| 29 | 66.59 | 81.70 | 82.81 |
| 30 | 66.40 | 80.90 | 82.71 |
| 31 | 66.62 | 82.11 | 82.59 |
| 32 | 66.36 | 81.84 | 83.07 |

Table 11: Training durations (s/epoch).

| Method | $T = 2$ | $T = 4$ | $T = 8$ | $T = 16$ | $T = 32$ |
|---|---|---|---|---|---|
| GPU=H20, batch size = 128 | | | | | |
| PSN | 10.07 | 16.79 | 27.47 | 50.27 | 102.42 |
| Autoselect ($k = 2$) | 12.63 | 22.25 | 40.99 | 84.65 | 153.95 |
| Autoselect ($k = 4$) | | 23.09 | 42.85 | 83.22 | 154.06 |
| Autoselect ($k = 8$) | | | 45.33 | 81.44 | 154.44 |
| GPU=A100, batch size = 32 | | | | | |
| PSN | 20.11 | 23.57 | 32.88 | 50.13 | 88.62 |
| Autoselect ($k = 2$) | 29.83 | 39.73 | 56.40 | 105.82 | 215.23 |
| Autoselect ($k = 4$) | | 35.11 | 50.91 | 104.40 | 225.88 |
| Autoselect ($k = 8$) | | | 52.17 | 96.41 | 187.18 |
| GPU=A100, batch size = 64 | | | | | |
| Autoselect | 13.72 | 17.46 | 27.47 | 45.84 | 85.79 |
| Autoselect ($k = 2$) | 19.75 | 31.94 | 47.13 | 96.79 | 197.29 |
| Autoselect ($k = 4$) | | 32.98 | 47.55 | 101.88 | 217.38 |
| Autoselect ($k = 8$) | | | 48.86 | 105.86 | 245.72 |

In Figure 3b, we mention that the gradient of Eq.(15) has jump points, which is detrimental to the network. On the sequence CIFAR dataset, we find that the network is still able to learn quite well. However, on the DVS-Lip dataset, as shown in Figure 7, using the original ste gradient causes the network to crash, resulting in the training and testing accuracy suddenly dropping to $1\%$. The reason is that the gradients appear to be the Not a Number (NaN) values.

# H  Limitations

Although we have designed multiple acceleration methods that significantly improve the training speed of the vanilla mul-free channel-wise PSN, its training speed is still slightly slower than that of PSN. This is mainly due to the additional overhead induced by quantization, and the memory read/write operations caused by *Vmap* during input/output processing in our SNNs being slower compared to dimension fusion. Nonetheless, the speed gap is not significant.

