# OpenReview forum: "Multiplication-Free Parallelizable Spiking Neurons with Efficient Spatio-Temporal Dynamics"
_NeurIPS.cc/2025/Conference — NeurIPS 2025 poster_

### Official Review · Reviewer_yzYg · 2025-06-20

**Clarity:** 3
**Significance:** 2
**Originality:** 2
**Rating:** 4
**Confidence:** 3

**Summary:**

This paper introduces a novel multiplication-free, channel-wise Parallel Spiking Neuron (PSN) model to improve the training efficiency of Spiking Neural Networks (SNNs), particularly in resource-constrained neuromorphic applications. Unlike conventional spiking neuron models that rely on serial iterative updates or costly high-order approximations, the proposed neuron model offers hardware-friendly acceleration by eliminating multiplications and enabling channel-level parallelism. The method is evaluated on neuromorphic audio (Spiking Heidelberg Digits), vision (DVS-Lip), and sequential image (CIFAR) datasets, achieving competitive or superior accuracy and demonstrating significant training speedups over prior SOTA neurons.

**Questions:**

1. What is the speedup of this new training method on embedded or low-end GPUs?

2. What is the accuracy of this new training method on other datasets such as imagenet?

**Ethical Concerns:**

["NO or VERY MINOR ethics concerns only"]

**Final Justification:**

I give a bordline accept. I am not the bottleneck to accept this paper.

**Limitations:**

yes

**Quality:**

3

**Strengths And Weaknesses:**

Strengths:

1. Important Problem Area:
The work targets a key bottleneck in neuromorphic computing—efficient training of SNNs—by addressing both latency and resource usage, which are crucial in embedded or real-time SNN deployment.

2. Hardware-Friendly Design:
The design eliminates multiplication operations, aligning well with the constraints of edge neuromorphic hardware and making it more suitable for low-power, high-efficiency environments.

Weaknesses:

1. Unclear Practical Impact of GPU Speedup:
The reported training speedups on a high-end A100 GPU may not convincingly reflect practical gains, especially given that CIFAR training is already fast on such hardware. Since the main selling point is hardware efficiency, evaluations on actual neuromorphic processors (e.g., Loihi, SpiNNaker) or low-resource edge devices would provide more meaningful insights.

2. Lack of Benchmark Diversity at Scale:
The paper does not include results on larger-scale datasets such as ImageNet. Demonstrating effectiveness at scale would further validate the method’s generalizability and robustness.

---

> ### Author Rebuttal · Authors · 2025-07-29
>
> > **W1**: Unclear Practical Impact of GPU Speedup: The reported training speedups on a high-end A100 GPU may not convincingly reflect practical gains, especially given that CIFAR training is already fast on such hardware. Since the main selling point is hardware efficiency, evaluations on actual neuromorphic processors (e.g., Loihi, SpiNNaker) or low-resource edge devices would provide more meaningful insights.
>
> We agree that evaluations on neuromorphic chips or edge devices would provide deeper insight into practical hardware efficiency. But due to the presence of time dimension T, training SNNs is significantly slower than ANNs, so improvements in GPU training speed can greatly facilitate research and development in this field.  Implementing and evaluating our method on neuromorphic hardware such as Loihi or SpiNNaker would require substantial engineering effort, and we have not yet completed such hardware-level verification. This remains an important direction for our future work.
>
> > **Q1**: What is the speedup of this new training method on embedded or low-end GPUs?
>
> While we have not yet conducted experiments on embedded or low-end GPUs due to hardware limitations, we evaluated the training speed of our method on an RTX 4090 GPU as a representative consumer GPU. As shown in Table R9, the speed gap between our autoselect implementation and PSN on the 4090 is similar to the results reported in the paper. Importantly, compared to the vanilla implementation, our method achieves up to a 10× speedup, demonstrating the effectiveness of our acceleration strategy. We believe these trends will generalize to resource-constrained GPUs.
>
> | Method \T       | 2     | 4      | 8      | 16     |
> | --------------- | ----- | ------ | ------ | ------ |
> | psn             | 12.16 | 14.20  | 25.76  | 49.84  |
> | Autoselect(k=2) | 18.80 | 23.40  | 47.18  | 83.27  |
> | Autoselect(k=4) |       | 24.86  | 47.52  | 86.81  |
> | Autoselect(k=8) |       |        | 44.57  | 85.05  |
> | Vannilla(k=2)   | 40.84 | 57.66  | 121.00 | 460.04 |
> | Vannilla(k=4)   |       | 407.09 | 439.61 | 576.29 |
> | Vannilla(k=8)   |       |        | 754.39 | 857.46 |
>
> Table R9: Training durations (s/epoch) of RTX 4090 GPUs on the CIFAR100 dataset.
>
> > **W2:** Lack of Benchmark Diversity at Scale: The paper does not include results on larger-scale datasets such as ImageNet. Demonstrating effectiveness at scale would further validate the method’s generalizability and robustness.
> >
> > **Q2**: What is the accuracy of this new training method on other datasets such as imagenet?
>
> We appreciate the reviewer's suggestion regarding evaluation on large-scale benchmarks such as ImageNet. In response, we conducted preliminary experiments using our method on the ImageNet dataset. Unfortunately, our approach did not outperform PSN under the default settings. Due to the limited rebuttal period, we did not further tune hyperparameters or conduct more extensive trials.
>
> Additionally, we observe that the Sliding PSN also underperforms PSN for static image classification tasks. We hypothesize that this may be attributed to the advantage of the standard PSN in accessing information from all global timesteps, which could be particularly beneficial in static image scenarios.

---

> > ### Comment · Reviewer_yzYg · 2025-08-01
> >
> > I have carefully reviewed the authors' rebuttal. While I appreciate their intention to address my comments in future work, I will maintain my original score. That said, I am not opposed to the acceptance of this paper and do not consider my evaluation to be a barrier to its acceptance.

---

> > > ### Author Response · Authors · 2025-08-08
> > >
> > > Thank you for your positive feedback on our work. We will carefully consider your suggestions and incorporate additional experiments and explanations in the manuscript of future revisions. We greatly appreciate your valuable input once again.

---

### Official Review · Reviewer_Hx7D · 2025-06-30

**Clarity:** 3
**Significance:** 3
**Originality:** 3
**Rating:** 5
**Confidence:** 4

**Summary:**

This paper proposes a novel spiking neuron model called the mul-free channel-wise Parallel Spiking Neuron, designed to improve the training speed and efficiency of SNNs while maintaining strong long-term dependency modeling.
Key innovations include the use of channel-wise convolution to improve representational capacity, sawtooth dilations to reduce model order (and thus complexity), and bit-shift operations to eliminate costly multiplications.
Experiments show superior performance compared to existing state-of-the-art spiking neurons, along with demonstrated improvements in training speed.

**Questions:**

Please refer to the weaknesses.

**Ethical Concerns:**

["NO or VERY MINOR ethics concerns only"]

**Final Justification:**

The authors have addressed my concerns. This paper is valuable for  parallelly training SNNs.

**Limitations:**

Yes.

**Quality:**

3

**Strengths And Weaknesses:**

## Strengths
1. I think it is important for the model to be explicitly designed to leverage GPU parallelism, addressing one of the long-standing challenges in SNN training.
2. Experiments are conducted on multiple benchmark datasets, showing broad applicability and state-of-the-art results.
3. The theory of the proposed method is complete.

## Weaknesses
I want to know:
1. Is the bit-shift replacement of multiplication universally applicable? Or does it introduce constraints on precision or dynamic range in practice, like FP-16, or other precisions?
2. How does the proposed model balance sparsity with dense operations like channel-wise convolution?
3. Has the proposed method been evaluated on real neuromorphic hardware (e.g., Loihi)? If not, how were the energy and latency benefits estimated?

---

> ### Author Rebuttal · Authors · 2025-07-29
>
> > W1: Is the bit-shift replacement of multiplication universally applicable? Or does it introduce constraints on precision or dynamic range in practice, like FP-16, or other precisions?
>
> For multiplication of IEEE 754-defined FP32/FP16 values with powers of 2, the operation can always be converted to adding a corresponding integer to the floating-point exponent bits [1]. For integers multiplied by powers of 2, the conversion is even simpler: direct bit shifts suffice.
>
> > W2: How does the proposed model balance sparsity with dense operations like channel-wise convolution?
>
> The sparsity in Spiking Neural Networks primarily refers to the sparse communication between layers. At the algorithm level, this means that neuron outputs are sparse binary spikes. At the hardware level, communication between computational cores (synapses + neurons) is made up of these sparse spikes, which can be efficiently implemented using sparse representation methods like Address-Event Representation (AER). However, the sparsity of neuronal dynamics itself is generally not constrained (for example, the calculation of membrane potential is typically dense).
>
> > W3: Has the proposed method been evaluated on real neuromorphic hardware (e.g., Loihi)? If not, how were the energy and latency benefits estimated?
>
> **Energy Estimation:**
>
> Deploying and verifying the proposed method directly on chips such as Loihi involves substantial engineering effort, and typically requires further steps including quantization of synaptic weights, which is beyond the current scope. Consistent with previous studies in the field [2-5], we estimate energy through the widely used energy estimation methodology from prior research in this field.
>
> Assume implementation on the 45nm CMOS technology, FP32 multiplication (MUL) and addition (ADD) operation consumes $ 3.7pJ$ and $0.9pJ$ of energy, respectively. By comparison, a shift (SHIFT) operation with 32-bit fixed-point (FIX32) data consumes $0.13pJ$ [2] and is faster [7].  **As we couldn't find the theortical energy of bit-shift operation of FP32, we use the FIX32 bit-shift energy for estimation, which maybe not necessarily completely precise**. By combining the neuron operation counts listed in Table 1 with the corresponding energy consumption for each operation, we can estimate the theoretical neuron energy consumption, as summarized in Table R7. Note that when the parameter $k$ reaches its maximum value $T$, the maximal theoretical energy cost of our neuron is given by $0.515T^2+1.415T$. For a large $T$, this is nearly $9\times$ lower than that of PSN, highlighting the substantial energy efficiency advantage of our approach in hardware-friendly scenarios.
>
> | Neuron | Operations                                                   | Energy (pJ)                          |
> | ------ | ------------------------------------------------------------ | ------------------------------------ |
> | PSN    | $(T^2+T)\times \text{ADD},T^2\times \text{MUL}$              | $4.6T^2+0.9T$                        |
> | Ours   | $((T+\frac{1-k}{2})\cdot k + T)\times \text{ADD}, ((T+\frac{1-k}{2})\cdot k)\times \text{SHIFT}$ | $1.03(T+\frac{1-k}{2})\cdot k +0.9T$ |
>
> Table R7:  Operations and energy of different spiking neurons during inference.
>
> Further, we use the sequential CIFAR100 Network structure `{{c128k3s1-BN-SN}*3-APK2s2}*2-FC256-SN-FC100` (see Appendix D for more details) to evaluate the energy comsuption on a single CIFAR100 Image. During inference, we could fuse the convolutional layer with the batch normalization layer and ignore the batchnorm layer. Follow [3], we use Eq.(R1) to calculate the energy of synaptic layers, where $\text{FL}^1_{\text{SNN Conv}}$ is the FLOPs of the first layer to encode static RGB images into spike-form, $N$ is the number of convolutional layers and $M$ is the number of fully connected layers.  $fr$ in Eq.(R2) is the firing rate of the input spike sequence of every synaptic layer, and here is the average value of the trained networkd across the entire test dataset. Assume the MAC and AC operations are implemented on the 45nm hardware, $E\_{MAC}=4.6pJ$ and $E\_{AC}=0.9pJ$. Following the experimental setup specified in Table 3, the neuron order $k$ of our neuron is $16$. Detailed Energy analysis is shown in Table R8, where the FLOPs and SOPs of the synaptic layers refer to the $\text{FL}\_{\text{SNN Conv}}^{1}$ and $\left(\sum\_{n=2}^N \text{SOP}^n_{\text{SNN Conv}} + \sum\_{m=1}^M{\text{SOP}^m\_{\text{SNN FC}}} \right)$ in the Eq.(R1), respectively. Please note that our energy consumption estimation takes into account the energy consumed by the computation of neurons themselves. In contrast, the majority of previous studies have only considered the energy consumption of synaptic layers while neglecting that of the neurons themselves.
>
> $$E_{\text{Synaptic}} = E\_{MAC}\times \text{FL}\_{\text{SNN Conv}}^{1}+E\_{AC} \times\left(\sum\_{n=2}^N \text{SOP}^n_{\text{SNN Conv}} + \sum\_{m=1}^M{\text{SOP}^m\_{\text{SNN FC}}} \right) \quad(\text{R1})$$
>
> $$\text{SOPs}(l) = fr\times T\times \text{FLOPs}(l)  \quad (\text{R2})$$
>
> |               | Neuron Layer                                             |                 | Synaptic Layer                                               |              | Total Energy$(\mu J)$ |
> | ------------- | -------------------------------------------------------- | --------------- | ------------------------------------------------------------ | ------------ | --------------------- |
> |               | Operations                                               | Energy$(\mu J)$ | Operations                                                   | Energy$(uJ)$ |                       |
> | $\text{PSN}$  | $1.91\times10^7 \text{MUL},1.97\times10^7 \text{ADD}$    | $88.56$         | $0.041\times 10^6 \text{FLOPs},3.194\times 10^6 \text{SOPs}$ | $3.06$       | $91.62$               |
> | $\text{Ours}$ | $7.32\times10^6 \text{SHIFT},7.92\times 10^6 \text{ADD}$ | $8.08$          | $0.041\times 10^6 \text{FLOPs},2.660\times 10^6 \text{SOPs}$ | $2.58$       | $10.66$               |
>
> Table R8:  Energy of our method and PSN on a single CIFAR100 Image.
>
> **Latency:**
>
> While both our approach and Sliding PSN require storing an input sequence proportional to neuron order $k$ for inference on neuromorphic chips, Sliding PSN typically needs a much larger $k$ to achieve stable performance for long-term dependencies. In contrast, our model achieves strong performance even with small $k$ (e.g., $k=2$), significantly reducing memory requirements for storing input sequences and making the solution more suitable for resource-constrained devices. **Details can be found in our reply to Reviewer ZuQM question 1, Table R2.**
>
> ```C++
> [1] Li, Xinlin, et al. "DenseShift: Towards Accurate and Efficient Low-Bit Power-of-Two Quantization." Proceedings of the IEEE/CVF International Conference on Computer Vision. 2023.
> [2] Huang Z, Fang W, Bu T, et al. Differential Coding for Training-Free ANN-to-SNN Conversion[C]//Forty-second International Conference on Machine Learning.
> [3] Zhou Z, Zhu Y, He C, et al. Spikformer: When Spiking Neural Network Meets Transformer[C]//The Eleventh International Conference on Learning Representations.
> [4] Rathi N, Roy K. Diet-snn: A low-latency spiking neural network with direct input encoding and leakage and threshold optimization[J]. IEEE Transactions on Neural Networks and Learning Systems, 2021, 34(6): 3174-3182.
> [5] Panda P, Aketi S A, Roy K. Toward scalable, efficient, and accurate deep spiking neural networks with backward residual connections, stochastic softmax, and hybridization[J]. Frontiers in Neuroscience, 2020, 14: 653.
> [6] You H, Li B, Huihong S, et al. ShiftAddNAS: Hardware-inspired search for more accurate and efficient neural networks[C]//International Conference on Machine Learning. PMLR, 2022: 25566-25580.
> [7] Elhoushi M, Chen Z, Shafiq F, et al. Deepshift: Towards multiplication-less neural networks[C]//Proceedings of the IEEE/CVF conference on computer vision and pattern recognition. 2021: 2359-2368.
> ```

---

> ### Comment · Reviewer_Hx7D · 2025-08-01
>
> The authors have addressed my concerns. I decide to raise my score.

---

> > ### Author Response · Authors · 2025-08-08
> >
> > Thank you for your positive feedback on our work. We will carefully consider your suggestions and incorporate additional experiments and explanations in the manuscript of future revisions. We greatly appreciate your valuable input once again.

---

### Official Review · Reviewer_DWxU · 2025-07-02

**Clarity:** 4
**Significance:** 3
**Originality:** 3
**Rating:** 4
**Confidence:** 3

**Summary:**

This paper redesigns the core neuron architecture of SNNs by combining per-channel separable convolution, zig-zag dilated convolution, and replacing multiplications with shift operations—augmented by an automated benchmarking system to select the optimal implementation - thereby achieving a multiplier-free design that efficiently captures long-range dependencies while remaining hardware-friendly. Experiments on SHD, sequential CIFAR, and DVS-Lip tasks reach or approach state-of-the-art performance, and training speed rivals that of traditional matrix multiplication.

**Questions:**

1.	By replacing all weights with power-of-two shifts, you remove multipliers—but real weights can be any real number. Have you quantified how much representational capacity you lose, especially for small or negative weights?
2.	How much memory do operations like reshape, depthwise + pointwise convolutions, and dilation actually consume in large models or with long sequences? The paper only reports training speed, but I believe training memory usage is what most researchers care about.
3.	Depth-wise + pointwise separation slashes parameters, but it also forces each channel’s temporal filter to be independent before mixing. Could that break important cross-channel correlations in the input spike train?
4.	In the Introduction, the authors claim that the method achieves significant energy savings, but the paper provides no analysis or experiments on energy consumption, nor is any such content found in the supplementary materials.
5.	While accelerating training is certainly important, for deployable methods acceleration during inference is even more critical. Have the authors evaluated whether this method’s inference speed offers sufficient advantages over PSN-based or sliding-PSN models with comparable parameter counts

**Ethical Concerns:**

["NO or VERY MINOR ethics concerns only"]

**Limitations:**

Yes.

**Paper Formatting Concerns:**

None.

**Quality:**

3

**Strengths And Weaknesses:**

Strengths:
The manuscript is well-written, logically rigorous, and clearly presented. This work makes a significant contribution to the field of parallel SNN neurons by avoiding expensive multiplications to accelerate and reduce the energy consumption of SNN computations, while also achieving competitively strong algorithmic performance.

Weaknesses:
Even though I think this paper is excellent, the method lacks evaluation on more advanced architectures such as Spikformer.

---

> ### Author Rebuttal · Authors · 2025-07-29
>
> > **W1**: Even though I think this paper is excellent, the method lacks evaluation on more advanced architectures such as Spikformer.
>
> We note that existing works related to parallel neurons [1-5] all do not conduct experiments on SNN-based Transformer architectures. Therefore, our work has also not conducted experiments on Spikformer-like architectures.
>
> > **Q1**: By replacing all weights with power-of-two shifts, you remove multipliers—but real weights can be any real number. Have you quantified how much representational capacity you lose, especially for small or negative weights?
>
> From the perspective of Taylor expansion, [6] suggests that small weights have a limited effect on network performance, which is also the theoretical foundation of pruning. Besides, to our knowledge, there is no theoretical analysis in the literature regarding the potential loss of representational capacity caused by power-of-two quantization of weights. Thus, we rely on empirical evidence to evaluate its impact. As shown in Table R3, on the sequential CIFAR100 dataset, we evaluate the same network with five different seeds (0-4). The test accuracy is $\pmb{67.2 \pm 0.1}$% before quantization, and $\pmb{66.5 \pm 0.2}$% after applying power-of-two quantization to the neuron weights. This suggests that quantization only results in a minor performance drop.
>
> | Datasets            | seed      | Ours (w/ Q)           | Ours (w/o Q)          |
> | ------------------- | --------- | --------------------- | --------------------- |
> | Sequential CIFAR100 | 0,1,2,3,4 | $\pmb{66.5 \pm 0.2}$% | $\pmb{67.2 \pm 0.1}$% |
>
> Table R3: Average test accuracy with $\pm$ standard deviation (%).
>
> > **Q2**: How much memory do operations like reshape, depthwise + pointwise convolutions, and dilation actually consume in large models or with long sequences? The paper only reports training speed, but I believe training memory usage is what most researchers care about.
>
> Table R4 presents the training memory usage of PSN and our methods (on the CIFAR100 dataset. "Ours (w/o Q)" uses channel-wise and dilated convolutions without quantization (Equation 10), while "Ours (w/ Q)" includes power-of-two quantization (Equation 13). The neuron order is $k=2$ for all cases.
>
> As $T$ increases, all methods require more memory. Compared to PSN, our methods use slightly more GPU memory for the same sequence length. For instance, when $T=16$, PSN uses 11,867 MB, while "Ours (w/o Q)" and "Ours (w/ Q)" require 13,529 MB and 16,361 MB, respectively.  **The reason why our method requires more training memory than PSN is that input padding is used to enable parallelized training.**
>
> | neuron\T    | 2    | 4    | 8    | 16    |
> | ----------- | ---- | ---- | ---- | ----- |
> | PSN         | 2101 | 3467 | 6315 | 11867 |
> | Ours(w/o Q) | 2507 | 3997 | 7333 | 13529 |
> | Ours(w/ Q)  | 2851 | 4835 | 8695 | 16361 |
>
> Table R4: Training memory (MB) on the training of the CIFAR100 dataset.
>
> Moreover, we would like to emphasize the significant practical advantage of our method during inference. Both our approach and Sliding PSN require storing an input sequence proportional to neuron order $k$. According to [4], Sliding PSN typically needs a large $k$ to achieve stable performance for long-term dependencies. In contrast, our neuron can converge with a much smaller $k$ and still achieve substantially better performance. For example, Sliding PSN could only achieve $62.11$% test accuracy with order $k=32$ on the sequential CIFAR100 dataset, while our method could attain $65.77$% test accuracy only under $k=4$ (Figure 5a, Appendix Table 7). Cause our approach dramatically reduces the required order $k$, it greatly reduces the length of the input sequence needs to be stored. As a result, our method requires much less memory during inference (Table R5, **547MB  vs. 2635MB, 4.8× reduction**).
>
> | neuron      | $k$  | Accuracy(%) | Memory (MB) |
> | ----------- | ---- | -------------- | ----------- |
> | Sliding PSN | 32   | 62.11          | 2635        |
> | Ours        | 4    | 65.77          | 547         |
>
> Table R5: **Step-by-step inference memory** on the sequential CIFAR100 ($T=32$) dataset.
>
> > **Q3**: Depth-wise + pointwise separation slashes parameters, but it also forces each channel’s temporal filter to be independent before mixing. Could that break important cross-channel correlations in the input spike train?
>
> The convolutional layers before neuron layers could effectively enable cross-channel information exchange, while the neuron layers aggregate temporal information within each channel. This division preserves cross-channel correlations rather than breaking them.
>
> > **Q4**: In the Introduction, the authors claim that the method achieves significant energy savings, but the paper provides no analysis or experiments on energy consumption, nor is any such content found in the supplementary materials.
>
> We have conducted a detailed analysis of energy consumption to support our claims regarding energy efficiency. Due to space limitations, **please refer to our response "Energy Estimation" to reviewer Hx7D' W3 for more details**. The analysis shows that our method achieves substantial savings compared to baseline models.
>
>
> > **Q5**: While accelerating training is certainly important, for deployable methods acceleration during inference is even more critical. Have the authors evaluated whether this method’s inference speed offers sufficient advantages over PSN-based or sliding-PSN models with comparable parameter counts
>
> As shown in Table R6, our method is slightly slower than PSN in inference on GPU. However, it is important to note that our design is primarily intended to minimize the use of multipliers for deployment on edge devices, rather than to optimize GPU inference speed. The key advantage of our approach is its hardware-friendliness—by replacing multiplications with bit-shift operations and reducing the demand of large neuron order $k$, these significantly reduces energy consumption and chip area, which is especially beneficial for resource-constrained scenarios.
>
> | neuron\T  | 2      | 4      | 8      | 16     | 32      |
> | --------- | ------ | ------ | ------ | ------ | ------- |
> | psn       | 0.8091 | 1.5430 | 2.8370 | 5.4668 | 10.7374 |
> | ours(k=2) | 1.0594 | 2.0243 | 3.4840 | 6.4227 | 12.5305 |
> | ours(k=4) |        | 2.0375 | 3.5812 | 6.5356 | 12.6378 |
> | ours(k=8) |        |        | 3.7819 | 6.7612 | 12.9210 |
>
> Table R6: Inference durations (s/epoch) of RTX 4090 GPUs on the CIFAR100 dataset.
>
> ```JavaScript
> [1] Feng W, Gao X, Du W, et al. Efficient Parallel Training Methods for Spiking Neural Networks with Constant Time Complexity[C]//Forty-second International Conference on Machine Learning.
> [2] Chen H, Yu L, Zhan S, et al. Time-independent Spiking Neuron via Membrane Potential Estimation for Efficient Spiking Neural Networks[C]//ICASSP 2025-2025 IEEE International Conference on Acoustics, Speech and Signal Processing (ICASSP). IEEE, 2025: 1-5.
> [3] Li Y, Sun Y, He X, et al. Parallel spiking unit for efficient training of spiking neural networks[C]//2024 International Joint Conference on Neural Networks (IJCNN). IEEE, 2024: 1-8.
> [4] Chen X, Wu J, Ma C, et al. Pmsn: A parallel multi-compartment spiking neuron for multi-scale temporal processing[J]. arXiv preprint arXiv:2408.14917, 2024.
> [5] Fang W, Yu Z, Zhou Z, et al. Parallel spiking neurons with high efficiency and ability to learn long-term dependencies[J]. Advances in Neural Information Processing Systems, 2023, 36: 53674-53687.
> [6] Li X, Liu B, Yang R H, et al. DenseShift: Towards Accurate and Efficient Low-Bit Power-of-Two Quantization[C]//Proceedings of the IEEE/CVF International Conference on Computer Vision. 2023: 17010-17020.
> ```

---

> ### Author Response · Authors · 2025-08-08
>
> Dear Reviewer DWxU,
>
> Thank you for your insightful comments. We hope that our responses have satisfactorily addressed your concerns.
>
> If you have any further questions, please feel free to contact us.

---

### Official Review · Reviewer_ZuQM · 2025-07-17

**Clarity:** 3
**Significance:** 3
**Originality:** 2
**Rating:** 4
**Confidence:** 1

**Summary:**

Traditional neuron models involve iterative step-by-step dynamics (implemented like RNN) which slow down computation. For some popular neuronal dynamics (like IF and LIF), one can express these dynamics in a way that allows for parallel computation as noted by Fang et al. [15]; in particular one can calculate how the membrane potential evolves over time without the resetting mechanism as in eq. (6) and calculate the binary spikes as in eq. (7) with learnable thresholds whose dimension is T (the number of time steps). The sliding PSN proposed by [15] uses shared weights of size k which convolve over the input sequence, allowing this model to use fewer weight parameters and handle variable length sequences (and not just T). This work builds on the sliding PSN of [15] by making the following changes: 1) first, this work uses a dilated convolution processing non-consecutive inputs and sets the dilation rates using a sawtooth wave-like heuristic (eq. (11)) and 2) replace multiplication with bit-shift operations and 3) quantize the the weights W. The authors also propose an autoselect algorithm (Algorithm 1) to choose amongst different acceleration options, which depend on data input shapes and GPUs. Finally, the authors apply these changes to SOTA SNN models for experimental validation, using the same architecture and hyperparameters as in [38] and [15] and benchmarking the results on a few datasets including SHD, (sequential) CIFAR10, CIFAR100, and DVS-Lip.

**Questions:**

In Figure 6, when comparing to PSN/Sliding PSN it is explained that this work is slower for T > 4. In the introduction, it is explained that the purpose of this work is to address the computational and memory limitations of sliding PSN. Could the authors perhaps comment more on these tradeoffs?

**Ethical Concerns:**

["NO or VERY MINOR ethics concerns only"]

**Final Justification:**

Reviewers have responded to my concerns and other reviewers' concerns with additional experiments which highlight the energy savings and memory-savings of the proposed method (at inference-time only, but that is the common use-case for SNN anyway). Although I am still a bit concerned that the work is not hugely novel, I think it can be a useful paper for practitioners to refer to (e.g., for using the code which will later be released, techniques for their own implementations).

**Limitations:**

yes

**Quality:**

3

**Strengths And Weaknesses:**

Strengths:
- The experiments seem to show SOTA performance on a few benchmarks (namely SHD, CIFAR10, CIFAR100).
- The method appears to be practical and (relatively) simple, which seems like it would be useful to many practitioners if the code/implementation were to be integrated into existing libraries.

Weaknesses:
- I feel that a lot of the ideas in this paper are not necessarily very “new” but pieces together many existing (known) techniques and put them together on top of the existing sliding PSN model/framework
- The justification for using the straight-through estimator for Q’(W) (and not considering any other methods) was a bit weak, especially considering the previous point. (STE is very simple to use is a first option to try.) Other methods for dealing with discontinuous and almost zero everywhere gradients exist, such as through using smooth surrogates.
- The improvements in test accuracy seem material but mild, and the uncertainty/error bars are not always reported for the authors’ work (I understand it may not always be available in other works, but having a sense of how tight the error bars are would help interpret some of the results), such as in Table 3 and 4.
- There doesn’t seem to be any indication that the code will be released publicly, which I think if done would strengthen the paper.

---

> ### Author Rebuttal · Authors · 2025-07-29
>
> > **S1:** ... it would be useful to many practitioners if the code/implementation were to be integrated into existing libraries.
> >
> > **W4:** There doesn’t seem to be any indication that the code will be released publicly, which I think if done would strengthen the paper.
>
> Thanks for your recognition. To ensure reproducibility and facilitate the review process, we have submitted our code as supplementary material, and the code will be made publicly available upon acceptance. Additionally, we plan to modularize the neuron implementation and contribute it to SpikingJelly [1] to make it easily accessible to the community.
>
> > **W1**: I feel that a lot of the ideas in this paper are not necessarily very “new” but pieces together many existing (known) techniques and put them together on top of the existing sliding PSN model/framework
>
> Our core contribution does not lie in inventing a wholly new, isolated component, but rather in the first systematic and non-trivial integration of several established techniques to address key bottlenecks in parallelizable SNN models such as Sliding PSN:
>
> 1. **Architecture – Resolving the Channel Mixing–Efficiency Trade-off:** A key limitation of PSN and Sliding PSN is their use of channel-shared neuron parameters, which restricts their ability to capture channel-specific features (see [2] Fig. S4). In contrast, our approach incorporates **channel-wise convolutions** into its neuronal dynamics. This design empowers each channel to learn independent temporal characteristics, thereby enhancing the model's representational power **without increasing FLOPs**. As demonstrated in Table 2, our method substantially outperforms both PSN and Sliding PSN, validating the effectiveness of this improvement.
> 2. **Efficiency – Resolving the Receptive Field–Resource Trade-off:** Sliding PSN requires a large neuron order $k$ to capture long-term dependencies, which limits its usability on resource-constrained hardware. We address this by introducing sawtooth dilations, which efficiently expand the temporal receptive field even with small $k$ (see Fig. 5b). Additionally, we employ bit-shift operations to replace costly multiplications in neuron dynamics, achieving substantial gains in hardware efficiency—reducing theoretical energy and area by $8×$(see lines 69–72).
> 3. **Practical Implementation – Bridging Theory and Real-world Efficiency**: We investigate multiple acceleration strategies that significantly improve the training speed of our neuorn in GPU environments. In spiking neural networks, synaptic layers primarily process spatial information, while neuron layers are tasked with modeling temporal dynamics. **Importantly, as  the neuron design becomes more complex, the memory layout challenges discussed in this paper tend to appear. The solutions and guidelines we present could serve as a valuable reference for a wide range of advanced and complex spiking neuron models.**
>
> In summary, our work systematically addresses the key challenges of performance and efficiency in the PSN model series. We introduce a fundamentally new parallel spiking neuron that achieves state-of-the-art accuracy across multiple datasets. Critically, our model is not only **highly efficient for GPU-based training** but also **optimized for deployment on resource-constrained edge devices**.
>
> > **W2:** The justification for using the straight-through estimator for Q’(W) (and not considering any other methods) was a bit weak, especially considering the previous point. (STE is very simple to use is a first option to try.) Other methods for dealing with discontinuous and almost zero everywhere gradients exist, such as through using smooth surrogates.
>
> For the round function, the most straightforward STE approach is as defined in Eq.(14). However, this leads to discontinuous gradients (see Fig. 3b), which can cause training to collapse (see Appendix F, Fig. S1). Therefore, we apply STE for Q'(W) to ensure stable training. Furthermore,  as shown in **Table R3**, the overall performance loss from power-of-two quantization and the STE surrogate gradient is minor and acceptable.
>
> > **W3**: The improvements in test accuracy seem material but mild, and the uncertainty/error bars are not always reported for the authors’ work (I understand it may not always be available in other works, but having a sense of how tight the error bars are would help interpret some of the results), such as in Table 3 and 4.
>
> **Regarding the magnitude of accuracy improvements:**
>
> We would like to highlight that the contribution of our work is best understood from two key perspectives.
>
> 1. Our primary objective is to **design a multiplication-free, hardware-friendly neuron**. By replacing costly floating-point multiplications with efficient bit-shift operations, our method achieves accuracy comparable to SOTA spiking neurons that rely on complex computations. This demonstrates a superior trade-off between accuracy and hardware cost.
> 2. More importantly, our model achieves **a significant performance improvement** over the most relevant baseline, Sliding PSN. As reported in Table 2, our method improves accuracy from **86.70%** to **91.17%** on Sequential CIFAR10, and from **62.11%** to **66.21%** on Sequential CIFAR100. These results provide strong evidence that our architectural optimizations result in substantial performance gains.
>
> **Regarding the stability of results and error bars:**
>
> In response to your suggestion, we have conducted additional statistical experiments on PSN and our method for the task in Table 3 (with five random seeds, 0–4), and on our method for the task in Table 4 (three seeds, 0–2). The results are presented in Table R1. Due to the high computational cost of training on DVS-Lip, we did not conduct extra runs for all baselines. As shown in Table R1, our method demonstrates very low deviation across runs, indicating strong robustness.
>
> | Datasets            | seed      | Baseline            | Ours             |
> | :------------------ | :-------- | :------------------ | :--------------- |
> | Sequential CIFAR100 | 0,1,2,3,4 | $61.79\pm0.47$([3]) | $66.5\pm0.2$     |
> | DVS-Lip             | 0,1,2     | $68.1$ ([4])        | $70.68 \pm 0.11$ |
>
> Table R1: Comparision of average test accuracy with $\pm$ standard deviation (%).
>
> > Q1: In Figure 6, when comparing to PSN/Sliding PSN it is explained that this work is slower for T > 4. In the introduction, it is explained that the purpose of this work is to address the computational and memory limitations of sliding PSN. Could the authors perhaps comment more on these tradeoffs?
>
> To achieve multiplication-free computation, during training we quantize the weights to power-of-two (details see Appendix C). This requires an additional forward and backward pass in each training step to reduce quantization error (while inference only needs one forward pass). Besides, practical implementations with operations such as vmap also incur certain memory read/write overhead, resulting in somewhat slower training compared to Sliding PSN. Importantly, the speed gap remains moderate and acceptable.
>
> However, we believe that these minor increases in training cost are justified by the substantial improvements in hardware efficiency and memory usage during inference.
>
> 1. On the hardware efficiency front, we replace multiplications in neuron dynamics with bit-shift operations, substantially improving hardware friendliness—this yields theoretical reductions in both energy and area consumption (see lines 69–72), addressing the **computational bottleneck** of Sliding PSN for edge deployment.
> 2. Regarding memory usage, during inference, both our approach and Sliding PSN require storing an input sequence proportional to neuron order $k$. According to [3], Sliding PSN typically needs a large $k$ to achieve stable performance for long-term dependencies. In contrast, our neuron can converge with a much smaller $k$ and still achieve substantially better performance. For example, Sliding PSN could only achieve $62.11\\%$ test accuracy with order $k=32$ on the sequential CIFAR100 dataset, while our method could attain $65.77\\%$ test accuracy only under $k=4$ (Figure 5a, Appendix Table 7). Cause our approach dramatically reduces the required order $k$, it greatly reduces the length of the input sequence needs to be stored. As a result, our method requires much less memory during inference (Table R2, **547MB  vs. 2635MB, 4.8× reduction**), significantly reducing **memory limitation** for storing input sequences and making the solution more suitable for resource-constrained devices.
>
> | neuron      | $k$  | Accuracy(%) | Memory (MB) |
> | ----------- | ---- | -------------- | ----------- |
> | Sliding PSN | 32   | 62.11          | 2635        |
> | Ours        | 4    | 65.77          | 547         |
>
> Table R2: **Step-by-step inference memory** on the sequential CIFAR100 ($T=32$) dataset.
>
> ```C++
> [1] Fang W, Chen Y, Ding J, et al. Spikingjelly: An open-source machine learning infrastructure platform for spike-based intelligence[J]. Science Advances, 2023, 9(40): eadi1480.
> [2] Fang W, Yu Z, Chen Y, et al. Incorporating learnable membrane time constant to enhance learning of spiking neural networks[C]//Proceedings of the IEEE/CVF international conference on computer vision. 2021: 2661-2671.
> [3] Fang W, Yu Z, Zhou Z, et al. Parallel spiking neurons with high efficiency and ability to learn long-term dependencies[J]. Advances in Neural Information Processing Systems, 2023, 36: 53674-53687.
> [4] Dampfhoffer M, Mesquida T. Neuromorphic lip-reading with signed spiking gated recurrent units[C]//Proceedings of the IEEE/CVF Conference on Computer Vision and Pattern Recognition. 2024: 2141-2151.
> ```

---

> > ### Comment · Reviewer_ZuQM · 2025-08-05
> >
> > Thank you to the authors for carefully addressing my comments/questions. My concerns have largely been addressed. After reading the other reviews and author response to those reviews, I have decided to raise my score. I hope that the authors can include these additional figures in the manuscript.

---

> > > ### Author Response · Authors · 2025-08-08
> > >
> > > Thank you for your positive feedback on our work. We will carefully consider your suggestions and incorporate additional experiments and explanations in the manuscript of future revisions. We greatly appreciate your valuable input once again.

---

### Note · Authors · 2025-08-14

We are grateful for the reviewers' valuable feedback and appreciate their recognition of our submission's key strengths.

- Achieves SOTA or highly competitive results on multiple benchmarks (ZuQM, DWxU, Hx7D).
- The multiplication-free design significantly reduces energy consumption and is well-suited for neuromorphic/edge hardware (DWxU, yzYg).
- The implementation explicitly considers GPU parallelism training, addressing a long-standing challenge in SNN training (Hx7D). The approach is practical and straightforward, valuable for other researchers (ZuQM).
- The manuscript is well-written and logically rigorous, supported by a complete theoretical foundation (DWxU, Hx7D).

The major concerns raised by the reviewers are as follows:

1. **Energy, Latency, and Memory (DWxU, Hx7D, ZuQM):** Reviewers inquire about the energy and latency characteristics and the trade-off between computation and memory.
   - **Energy:** We provide a detailed energy analysis in Tables R7 and R8. These results demonstrate that our method achieves substantial energy savings, consuming nearly **9× less energy** than PSN in end-to-end inference on CIFAR100.
   - **Latency & Memory:** Table R5 details the inference memory footprint. As our method achieves strong performance with a much lower neuron order, it requires significantly less memory, which in turn contributes to reduced latency.
2. **Training and Inference Speed (yzYg, DWxU):** Reviewers also raise questions regarding the training and inference speed on commodity GPUs.
   - Our benchmarks on an RTX 4090 (Tables R6 and R9) show that our method is modestly slower than PSN. However, we argue this slight speed trade-off is well-justified by the compelling advantages of our approach: a **multiplication-free design, lower neuron order, high accuracy, and significantly lower energy consumption**.
3. **Impact of Quantization (ZuQM, DWxU):** Reviewers are interested in the accuracy impact of our power-of-two quantization.
   - As presented in Table R3, our method maintains robust performance, with the quantization step introducing a minimal accuracy drop of only **0.7 ± 0.1%**.

We are happy to have addressed the concerns of reviewers **ZuQM** and **Hx7D**, and we thank them for increasing their scores. We also appreciate the positive ratings from reviewers **yzYg** and **DWxU**. The clarifications and additional results from this rebuttal will be incorporated into the revised manuscript.

---

### Decision · Program_Chairs · 2025-09-17

**Decision:**

Accept (poster)

**Comment:**

The authors propose a multiplication-free framework for parallelizable spiking neural networks. Their approach improves state-of-the-art approaches in this area with changes to the architecture by per-channel separable convolutions, bit shifts instead of multiplications, and saw-tooth dilation for convolutions. The reviewers remark that none of these changes by themselves are by themselves necessarily revolutionary. However, they acknowledge that the combination leads to a system with state-of-the-art accuracy on common SNN tasks, while getting rid of multiplications, which is likely to considerably reduce energy expenditure.

The authors successfully addressed reviewer questions on the variability of the results, energy and memory consumption, the cost of quantization, and the memory comparison with PSN.

The work would be even more convincing when the expected energy gains would be shown on neuromorphic hardware, and if the accuracy would extend to more complex tasks. However, I agree with the authors that these elements can be addressed in future work. This article will definitely be of interest to practitioners in the field.